# Instructing Goal-Conditioned Reinforcement Learning Agents with Temporal Logic Objectives

**Wenjie Qiu**[*]
Rutgers University
wq37@cs.rutgers.edu

**Wensen Mao**[*]
Rutgers University
wm300@cs.rutgers.edu

**He Zhu**
Rutgers University
hz375@cs.rutgers.edu

## Abstract

Goal-conditioned reinforcement learning (RL) is a powerful approach for learning general-purpose skills by reaching diverse goals. However, it has limitations when it comes to task-conditioned policies, where goals are specified by temporally extended instructions written in the Linear Temporal Logic (LTL) formal language. Existing approaches for finding LTL-satisfying policies rely on sampling a large set of LTL instructions during training to adapt to unseen tasks at inference time. However, these approaches do not guarantee generalization to out-of-distribution LTL objectives, which may have increased complexity. In this paper, we propose a novel approach to address this challenge. We show that simple goal-conditioned RL agents can be instructed to follow arbitrary LTL specifications without additional training over the LTL task space. Unlike existing approaches that focus on LTL specifications expressible as regular expressions, our technique is unrestricted and generalizes to $\omega$-regular expressions. Experiment results demonstrate the effectiveness of our approach in adapting goal-conditioned RL agents to satisfy complex temporal logic task specifications zero-shot.

## 1 Introduction

Goal-conditioned learning Liu et al. [2022] is a type of reinforcement learning (RL) task where an agent learns to achieve a specific goal in a given environment. The goal can be defined in various ways, such as a particular state of the environment or a desired outcome, and the agent learns to map its current observations to actions that bring it closer to the goal. By providing a high-level goal, the agent can effectively learn to generalize its behavior and adapt to changing conditions.

However, goal-conditioned RL agents struggle with generalizing to more complex, task-oriented policies when the goals are specified by temporally extended instructions typically written in Linear Temporal Logic (LTL) languages, which creates a much larger task space than simple goals. In such cases, as the LTL task space is vast and difficult to exhaustively sample, it becomes more challenging for the agent to identify the correct sequence of actions needed to accomplish a task, which results in slow learning and poor generalization.

We propose a novel approach that enables simple goal-conditioned RL agents to follow arbitrary LTL specifications without any additional training over the LTL task space. This is in sharp contrast to existing approaches for finding LTL-satisfying policies, which require sampling a large set of LTL instructions from the task space during training as "goals" Vaezipoor et al. [2021], Kuo et al. [2020]. Such methods have no guarantee in generalizing to unseen tasks outside their training distribution of LTL objectives. As an example, the capability of most existing learning algorithms for LTL-satisfying policies is restricted to specifications over regular expressions that can be modeled by reward machines Icarte et al. [2022, 2018]. Our technique can handle $\omega$-regular LTL specifications

---

[*]Wenjie Qiu and Wensen Mao contributed equally to this work.

37th Conference on Neural Information Processing Systems (NeurIPS 2023).

over infinite control sequences even though the underlying goal-conditioned agents have never seen such specifications during training.

Our main contribution lies in a policy learning algorithm for LTL satisfaction, which decouples complex low-level environment interaction from high-level task planning. Specifically, we leverage a goal-conditioned RL agent to interact with low-level environments to learn how to achieve basic goals without considering higher-level tasks. The goal-conditioned RL agent has the capability of measuring the difficulty of transitioning between goals in the goal space. To solve an unseen LTL instruction $\varphi$, our method applies a weighted graph search algorithm that engages in high-level planning over the difficulty of achieving the specified goals to choose the optimal sequence of sub-goals in $\varphi$ for the goal-conditioned agent to reach. We present experimental results demonstrating the effectiveness of our technique in adapting goal-conditioned RL agents to satisfy complex LTL task specifications of varying complexity zero-shot.

## 2 Background

**Goal-Augmented Markov Decision Processes (MDPs)** extend standard MDPs by incorporating a set of goals $\mathcal{G}$ within the system state space $S$. We assume the existence of a state-goal labeling function $L : S \rightarrow 2^{\mathcal{G}}$ that maps each state to a set of atomic propositional symbols used to describe the observation of a system state where an atomic propositional symbol $g \in \mathcal{G}$ is a variable that takes on a truth. Throughout the paper, the terms "goals" and "propositions" are used interchangeably.

As an example, consider ZoneEnv, a Safety Gym environment Achiam and Amodei [2019] adapted from Vaezipoor et al. [2021]. The environment (Fig. 1) is a walled 2D plane with colored zones that correspond to task propositions. Both the zones and the point robot are randomly positioned on the plane. In this example, we use a set of atomic propositions $\mathcal{G} = \{y, w, r, j\}$ that represents the four colors yellow, white, red, and jet black.

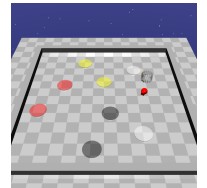 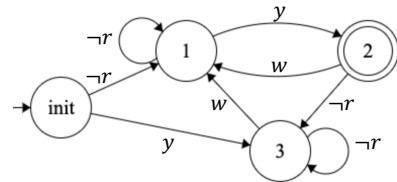

Figure 1: An LTL Task for ZoneEnv.

Formally, we model a goal-augmented MDP as a tuple $(S, A, \mathcal{G}, L, T, R, \gamma)$ that consists of a state space $S$, an action space $A$, a goal space defined by a set of atomic propositions $\mathcal{G}$, a transition function $T : S \times A \times S \rightarrow [0, 1]$ that maps each state-action-state transition to a probability, a labeling function $L$, and a reward function defined with the goals $R : S \times \mathcal{G} \times A \rightarrow \mathbb{R}$ that maps each state-action transition to a scalar reward for a particular goal in $\mathcal{G}$. $\gamma \in [0, 1)$ is a discount factor that controls the relative importance of immediate versus future rewards.

In a goal-augmented MDP, the agent's objective is to reach a goal via a goal-conditioned policy $\pi : S \times G \times A \rightarrow [0, 1]$ that maximizes the expectation of the cumulative reward: $J(\pi) = \mathbb{E}_{\substack{a_t \sim \pi(\cdot|s_t, g), g \sim P_g \\ s_{t+1} \sim T(\cdot|s_t, a_t)}} \left[ \sum_{t=0}^{\infty} \gamma^t R(s_t, g_t, a_t) \right]$ where $p_g$ be a distribution over goals in $G$. The optimal policy $\pi^* = \arg\max_\pi J(\pi)$ maximizes the expected cumulative return over all possible goals, weighted by their probability under the goal distribution $P_g$. This formulation encourages the policy to be effective at achieving a diverse set of goals, rather than simply optimizing for a single goal. For example, in the colored zone environment in Fig. 1 (left), a goal-conditioned agent policy $\pi(a|s, g)$ can be trained to reach a specific colored zone $g \in \mathcal{G} = \{y, w, r, j\}$.

The value function for goal-conditioned reinforcement learning $V^\pi(s_t, g)$ returns the expected cumulative discounted reward obtained by following policy $\pi$ starting from state $s_t$ with the goal $g$ as the target: $V^\pi(s_t, g) = \mathbb{E}_{\substack{a_t \sim \pi(\cdot|s_t, g), \\ s_{t+1} \sim T(\cdot|s_t, a_t)}} \left[ \sum_{t=0}^{\infty} \gamma^t R(s_t, g, a_t) \right]$ The goal-conditioned Q-function $Q^\pi(s_t, g, a)$ returns the expected cumulative discounted reward obtained by taking action $a$ in state $s_t$, and then following policy $\pi$ thereafter, with the goal $g$ as the target: $Q^\pi(s_t, g, a) = r(s_t, g, a) + \gamma \sum_{s_{t+1} \in S} T(s_{t+1}|s_t, a) V^\pi(s_{t+1}, g)$ The goal-conditioned $Q$ and $V$ functions may be learned using model-free RL algorithms such as $Q$-learning or policy gradient methods. In training, we assume goals can be uniformly sampled. We also assume that $t_{max}$ is the maximum horizon and a goal-

conditioned agent receives a positive reward signal only when it successfully achieves the specified goal by the end of a training episode by framing a reward function:

$$r(s_t, g, a_t) = \mathbb{1}[t = t_{max}] \cdot \mathbb{1}[g \in L(s_t)] \tag{1}$$

where $\mathbb{1}$ is the indicator function. This reward function has the following interpretation: given a state $s_t$, will the goal-conditioned policy $\pi$ get to a goal $g$ after $t$ time steps of attempting to reach $g$.

**Linear Temporal Logic (LTL)** is a formal language used to specify the temporal properties of a system. LTL formulas are built from a set of temporal operators, such as $\mathbf{X}$ ("next"), $\mathbf{F}$ ("eventually"), $\mathbf{G}$ ("always"), and $\mathbf{U}$ ("until"). LTL formulas are constructed over a finite set of atomic proposition symbols $\mathcal{G}$. Formally, the syntax of LTL formulas is defined recursively as follows:

$$\varphi ::= g \mid \neg\varphi \mid \varphi \wedge \varphi \mid \varphi \vee \varphi \mid \mathbf{X}\varphi \mid \varphi\mathbf{U}\varphi \mid \mathbf{F}\varphi \mid \mathbf{G}\varphi \quad \text{where } g \in \mathcal{G}$$

Intuitively, the formula $\mathbf{X}\varphi$ holds if $\varphi$ holds at the next time step, $\varphi_1\mathbf{U}\varphi_2$ holds if $\varphi_1$ holds until $\varphi_2$ holds, $\mathbf{F}\varphi$ holds if $\varphi$ holds at a future step, and $\mathbf{G}\varphi$ holds if $\varphi$ holds at the current and all future steps.

An LTL formula $\varphi$ is said to be satisfied by a state transition system with state space $S$ if and only if all infinite sequences of states of the system satisfy $\varphi$. We assume the existence of a labeling function $L : S \to 2^{\mathcal{G}}$ that maps each state to the atomic propositions that are true in that state. Given an infinite sequence of system states $\tau = \langle s_0, s_1, s_2, \ldots \rangle$ generated by a system, $\tau \vDash \varphi$ models the satisfaction relation $\vDash$ between an LTL formula $\varphi$ and the sequence of system states $\tau$. Formally, $\tau \vDash \varphi$ if and only if $s_0 \vDash \varphi$ which. We can define $s_i \vDash \varphi$ inductively, as illustrated on the right.

- $s_i \vDash g$ if $g \in \mathcal{G}$ is an atomic proposition and $g \in L(s_i)$ holds.
- $s_i \vDash \neg\varphi$ if $s_i \nvDash \varphi$.
- $s_i \vDash \varphi_1 \wedge \varphi_2$ if $s_i \vDash \varphi_1$ and $s_i \vDash \varphi_2$.
- $s_i \vDash \varphi_1 \vee \varphi_2$ if $s_i \vDash \varphi_1$ or $s_i \vDash \varphi_2$.
- $s_i \vDash \mathbf{X}\varphi$ if $s_{i+1} \vDash \varphi$.
- $s_i \vDash \mathbf{F}\varphi$ if there exists $j \geq i$ such that $s_j \vDash \varphi$.
- $s_i \vDash \mathbf{G}\varphi$ if for all $j \geq i$, $s_j \vDash \varphi$.
- $s_i \vDash \varphi_1\mathbf{U}\varphi_2$ if there exists $j \geq i$ such that $s_j \vDash \varphi_2$, and for all $i \leq k < j$, $s_k \vDash \varphi_1$.

In the ZoneEnv example in Fig. 1 (left), the robot is tasked to oscillate between yellow zones and white zones while always avoiding red zones. Assume a set of atomic propositions $\mathcal{G} = \{y, w, r, j\}$ that represents the four colors yellow, white, red, and jet black. The LTL objective for this task can be specified as:

$$\mathbf{GF}(y \wedge \mathbf{XF}w) \wedge \mathbf{G}(\neg r) \tag{2}$$

**Büchi Automaton** is a type of automaton used to recognize languages that consist of infinite sequences of symbols. A Büchi automaton is defined by a tuple $(Q, \Sigma, \delta, q_0, F)$, where $Q$ is a finite set of automaton states, $\Sigma$ is a finite alphabet of symbols, $\delta : Q \times \Sigma \to 2^Q$ is a transition function that maps each state and symbol to a set of states, $q_0 \in Q$ is the initial state, and $F \subseteq Q$ is a set of accepting states. Informally, a Büchi automaton recognizes a language consisting of all infinite sequences of symbols (corresponding to automaton runs) that visit an accepting state infinitely often.

**Büchi Automaton Construction from LTL**. Büchi automata can be used to represent temporal specifications in LTL. Given an LTL formula $\varphi$ over a set of propositional symbols in $\mathcal{G}$, a Büchi automaton $\mathcal{B}$ can be constructed that recognizes all infinite sequences of system states that satisfy $\varphi$. The alphabet $\Sigma$ of the Büchi automaton $\mathcal{B}$ is over the set of atomic propositions $\mathcal{G}$ of $\varphi$, i.e., $\Sigma = 2^{\mathcal{G}}$. That is, an infinite sequence of system states $s_0, s_1, s_2, \ldots$ satisfies $\varphi$ if and only if there exists an infinite sequence of automaton states $q_0, q_1, q_2, \ldots$ in $\mathcal{B}$ such that $q_0$ is the initial automaton state, for all $i \geq 0$, if $s_i \vDash w_i$ where $w_i \in \Sigma$ then $\delta(q_i, w_i)$ contains at least one state, and the set of states that occur infinitely often in the sequence $q_0, q_1, q_2, \ldots$ is a subset of the accepting states of $\mathcal{B}$.

For example, for the ZoneEnv task in Fig. 1 (left), the converted Büchi automaton for the LTL task in Eq. 2 is depicted in Fig. 1 (right). The only accepting state in this machine is state 2. To reach the accepting state, a yellow zone must be visited first. The Büchi automaton accepts when state 2 is reached infinitely often, meaning that the agent must always make a loop back to state 2 from state 3 and then state 1 by visiting a white zone and a yellow zone. This kind of infinite looping behavior cannot be expressed by a regular expression.

# 3 Instructing Goal-Conditioned Agents with LTL Objectives

**Task-Augmented Markov Decision Processes**. We now formalize the problem of instructing an RL agent to follow LTL objectives. To this end, we generalize Goal-Augmented MDPs to Task-Augmented MDPs $(S, A, \mathcal{G}, \Phi, L, T)$ where the definitions for state space $S$, action space $A$, goal space $G$, labeling function $L$ (from states to atomic propositions in $\mathcal{G}$), state transition function $T$ remain the same. $\Phi$ is the universe space of LTL formulas that can be constructed from the propositions in $\mathcal{G}$. We would like to construct a policy $\pi(a|s, \varphi)$ such that for any possible LTL formula $\varphi \in \Phi$, the policy $\pi(a|s, \varphi)$ has the highest probability of LTL specification satisfaction with respect to $\varphi$:

$$\pi^* = \arg\max_{\pi} \mathbb{E}_{\tau \sim \pi(\cdot|\cdot, \varphi)} \left[ \mathbb{1}[\tau \vDash \varphi] \right] \tag{3}$$

In this formulation, a task-augmented MDP does not have a reward function. Instead, the objective is to generate the most number of LTL-satisfying runs for a given LTL task. The MDP does not have a maximum horizon as the agent is expected to run over an infinite time horizon to satisfy an ($\omega$-regular) LTL instruction (e.g., Fig. 1).

Solving task-augmented MDPs turns out to be significantly more challenging than learning for goal-augmented MDPs due to the extremely large task space, which grows exponentially with the number of goals. The agent must identify the correct sequence of sub-goals to accomplish an LTL task. For the ZoneEnv example in Fig. 1, an agent may be instructed to accomplish a task by reaching a white zone first and then proceeding to a red zone in two distinct phases. In each phase, the zone to avoid changes (avoiding jet black and then yellow). We specify this task in Eq. 4 whose Büchi automaton representation is depicted in Fig. 2.

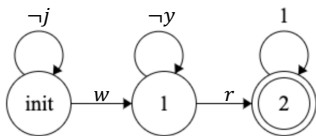

Figure 2: Another LTL task for ZoneEnv.

$$\neg j \mathbf{U}(w \wedge (\neg y \mathbf{U} r)) \tag{4}$$

The two LTL tasks in Eq. 2 and Eq. 4 express very different requirements and require distinct logic to solve. The agent is expected to successfully fulfill both tasks during inference.

## 3.1 LTL Task Planning on Büchi Automata

Our approach relies on the principle that an accepting run of a Büchi automaton must contain an accepting state that is reachable from the initial state and lies on a cycle (e.g., Fig. 1). This means that accepting runs of a Büchi automaton can be searched on top of a directed graph representation of the automaton. A goal-conditioned agent can then be used to subsequently achieve the goals along the search paths for task execution.

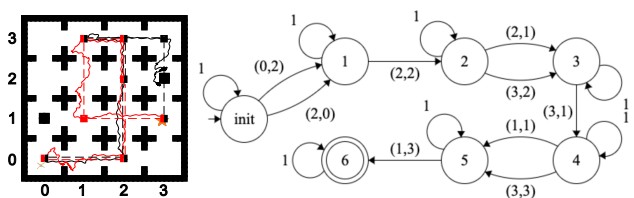

Figure 3: An LTL Task for Ant-16rooms where the red path is more feasible than the black path.

However, this simple strategy may yield suboptimal task performance. Consider a Mujoco ant navigation task in a 16-room environment depicted in Fig. 3(left). The rooms are separated by thick walls and are connected through bi-directional doors. The agent is initially positioned at the center of the bottom-left room and is given the following LTL instruction to traverse through the centers of a series of rooms with choices:

$$\mathbf{F}(((0,2) \vee (2,0)) \wedge \mathbf{F}((2,2) \wedge \mathbf{F}(((2,1) \vee (3,2)) \wedge \mathbf{F}((3,1) \wedge \mathbf{F}(((1,1) \vee (3,3)) \wedge \mathbf{F}(1,3))))))$$

Here an atomic proposition $g \in \mathcal{G}$ is in the form of $(r, c)$ denoting the room in the $r$-th row and $c$-th column. The bottom-left corner is room $(0, 0)$. Given an environment state $s$, define $s \vDash (r, c)$ is true if the position of the Mujoco ant in $s$ is close to the center of the $(r, c)$-th room within a threshold. The Büchi automaton $\mathcal{B}$ converted from the LTL instruction is given in Fig. 3(right). There are 8 paths on $\mathcal{B}$ to reach the accepting state 6. A graph search algorithm could yield any of these paths for the Mujoco ant to traverse along the chosen route. However, not all of them are equally optimal.

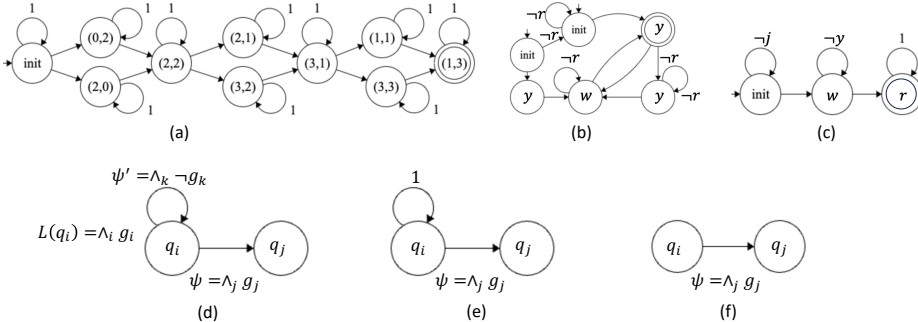

Figure 4: Graphs (a), (b), and (c) are converted from the Büchi automata in Fig. 3, Fig. 1, and Fig. 2. (d), (e), and (f) are types of transitions in a graph representation of a Büchi automaton.

Some of the routes require the ant to take a detour to avoid the black obstacles placed in certain room centers. In the worst case, the agent may be blocked by the obstacles and cannot complete the task.

In order to optimally accomplish the sequences of tasks defined in an LTL instruction $\varphi$ with atomic propositions in $\mathcal{G}$, we perform high-level reasoning by evaluating the difficulty of achieving the specified goals (i.e. atomic propositions) in $\varphi$ in the goal space $\mathcal{G}$. Our technique exploits the value function $V^\pi(s, g)$ of a goal-conditioned agent $\pi(\cdot|s, g)$ to capture the agent's transition capabilities. For any pair of goals, we learn a *goal value function* $\mathcal{V}^\pi : S \times \mathcal{G} \times \mathcal{G} \to \mathbb{R}$ to estimate the capability of taking the goal-conditioned policy $\pi$ to go from one goal to another goal on average. The function is parameterized by an environment state $s$ because the agent's capability to transition between goals can differ depending on various environment layouts and agent states. To get the desired estimate, we regress $\mathcal{V}$ towards the value function of $\pi$ by minimizing:

$$\min_{\mathcal{V}} \left( V(s_t, g_k) - \mathcal{V}(s_j, g_t, g_k) \right)^2 \quad \text{where } g_k \in L(s_k) \wedge g_t \in L(s_t) \tag{5}$$

with $\tau \sim B$ (a replay buffer), $t \sim \{0 \ldots t_{max}\}$, $s_t, a_t, s_{t+1} \sim \tau$, $j \sim \{0, t\}$, $k \sim \{t+1 \ldots t_{max}\}$, and $L$ as the state labeling function. We train $\mathcal{V}$ together with a goal-conditioned learning algorithm.

With the learned goal value function $\mathcal{V}$ of a goal-conditioned agent as a measurement to assess its capability to transition between goals, we reduce the task of learning an LTL specification satisfying policy to finding the optimal accepting run in the Büchi automaton representation of the LTL specification. We convert a Büchi automaton $\mathcal{B} = (Q, \Sigma, \delta, q_0, F)$ as a directed graph $G_{\mathcal{B}} = (\mathcal{Q}, \Sigma, \Delta, \mathcal{Q}_0, \mathcal{F}, \mathcal{L})$ with nodes $\mathcal{Q}$, node transitions (edges) $\Delta$ over the same alphabet $\Sigma$, a set of initial nodes $\mathcal{Q}_0$ and accepting nodes $\mathcal{F}$. The conversion from $\mathcal{B}$ to graph $G_{\mathcal{B}}$ starts with setting $\mathcal{Q} = Q$, $\Delta = \delta$, $\mathcal{Q}_0 = \{q_0\}$, and $\mathcal{F} = F$. The node labeling function $\mathcal{L}$ maps nodes in $\mathcal{Q}$ to atomic propositions (or goals) in the alphabet $\Sigma$. Intuitively, $\mathcal{L}$ approximates the specific region in the goal space where the agent should be positioned at a given node. For each *non-self* transition $(q_i, \psi, q_j) \in \delta$ in $\mathcal{B}$ where $\psi$ encodes the goal space to which the transition is enabled, we have $\mathcal{L}(q_j) = \psi$. As an example, Fig. 4(a) depicts the converted graph representation $G_{\mathcal{B}}$ from the Büchi automaton in Fig. 3(right). For the transition from the initial node to node 1, the node labeling function $\mathcal{L}$ maps node 1 to goal room $(0, 2)$. This transition denotes that the agent reaches room $(0, 2)$ from the initial room. If there are several incoming transitions with different alphabet symbols to $q_j$ in $\mathcal{B}$, we duplicate $q_j$ and its outgoing transitions in $G_{\mathcal{B}}$ so that each duplicated node for $q_j$ captures one of the possible transitions to $q_j$. For example, in Fig. 3, there are two non-self Büchi automaton transitions to state 1 which are encoded as node transitions from the initial room to room $(0, 2)$ and $(2, 0)$ respectively in Fig. 4(a). An exception for constructing the node labeling function $\mathcal{L}$ occurs when $\psi$ takes the form of $\wedge_k \neg g_k$ on a non-self transition $(q_i, \psi, q_j)$ where all atomic propositions $g_k$ are negative, indicating a specific area in the goal space that must be avoided to activate the transition. In this case, the designated goal area outlined by $\psi$ for $q_j$ can be excessively large. We instead assign $\mathcal{L}(q_j) = \mathcal{L}(q_i)$. Intuitively, if $g_k$ does not intersect with the current state on $\mathcal{L}(q_i)$, the agent can trivially fulfill the avoidance requirement by making a "jumped" transition to $q_j$. For example, the "avoidance" transition from state 2 to state 3 on the Büchi automaton in Fig. 1(right) is encoded as the rightmost non-self transition in the graph representation of the automaton in Fig. 4(b).

We interpret the semantics of a graph representation $G_{\mathcal{B}}$ converted from a Büchi automaton $\mathcal{B}$ through the different kinds of *non-self* transitions that $G_{\mathcal{B}}$ may have, as visualized in Fig. 4(d), (e), and (f). In

Fig. 4(d), on a non-self transition $(q_i, \psi, q_j)$, in order to transit to $q_j$, the goal-conditioned agent is tasked to reach the goal region $\psi \equiv \bigwedge_j g_j$ from the goal region $\mathcal{L}(q_i) \equiv \bigwedge_i g_i$. The "self-transition" on a node $q_i$ describes a goal-related constraint $\psi' = \bigwedge_k \neg g_k$ that must be maintained until the goal-conditioned agent can transit to $q_j$. The agent can take an unlimited number of steps to reach the goal region $\psi$, provided that each step avoids entering any undesired region $g_k$ within $\psi'$. For weighted graph search, we assign the weight of such a transition as:

$$\mathcal{W}(s, (q_i, \psi, q_j)) = \max_{g_i, g_j} -\log \mathcal{V}(s, g_i, g_j) \text{ where } \mathcal{L}(q_i) = \bigwedge_i g_i \text{ and } \psi = \bigwedge_j g_j \tag{6}$$

The weight encodes the capability of a goal-condition agent to transit from $q_i$ to $q_j$ from an observation at environment state $s$. In Fig. 4(d), we restrict $\mathcal{L}(q_i)$, $\psi$, and $\psi'$ to conjunctions of atomic propositions or negated propositions. In Appendix G.3, we extend this definition to include formulas as conjunctions of propositions and their negations to support more complex LTL tasks, such as $\mathbf{F}(g_1 \wedge \neg g_2)$.

In Fig. 4(e), the transition is similar to that of Fig. 4(d), and the agent can also take as many steps as necessary to reach the goal region specified by $\psi$ by utilizing the self transition on the source node $q_i$. However, there are no specific avoidance requirements to consider. We use Equation 6 to weigh this kind of transition. In Fig. 4(f), the goal-conditioned agent is required to reach the goal region $\psi$ in exactly *one* step as there is not any self transition that can be taken upon $q_i$. We set $\mathcal{W}(s, (q_i, \psi, q_j)) = 0$ (the full capacity) if $\mathcal{L}(q_i) \Leftrightarrow \psi$ i.e. the agent can directly "jump" to $q_j$ (e.g. see the rightmost non-self transition in Fig. 4(b)). Otherwise, we assign it the lowest capacity $\mathcal{W}(s, (q_i, \psi, q_j)) = \infty$. This is because our approach cannot explicitly pick an action to reach a goal region in a single step (we instead depend on the goal-conditioned agent to take as many steps as needed for goal reaching).

## 3.2 Algorithm Summary

Our technique generates policies for a LTL specification $\varphi$ based on a goal-conditioned RL agent $\pi$ and a learned goal value function $\mathcal{V}$ as follows:

**1.** We first convert $\varphi$ to a Büchi automaton $\mathcal{B}$, which is subsequently converted to a graph representation $G_{\mathcal{B}}$ using the technique illustrated in Sec. 3.1.

**2.** Associate each transition $(q_i, \psi, q_j)$ on $G_{\mathcal{B}}$ with weight $\mathcal{W}(s, (q_i, \psi, q_j))$ according to Equation 6 to measure the capability of the goal-conditioned agent to make the transition. In our implementation, we set $s$ as the initial state of an episode. A more advanced planning strategy may conduct task planning every $h$ timesteps from a current environment state for the remaining sub-goals on $G_{\mathcal{B}}$ and update the weight of any transition based on the current state before replanning.

**3.** Decompose $G_{\mathcal{B}}$ into strongly connected components (SCCs) using Tarjan's algorithm Tarjan [1972].

**4.** To find an optimal path on $G_{\mathcal{B}}$ for task execution to satisfy the LTL $\varphi$, we follow these steps:

- For each accepting state $s_a$ within a maximal SCC Geldenhuys and Valmari [2004], we use Dijkstra's algorithm to find the shortest path from the initial state to $s_a$, denoted as $p$.
- Next, we apply Dijkstra's algorithm to find the shortest cycle from $s_a$ back to $s_a$ in the maximal SCC, denoted as $q$.
- The optimal path for the accepting state $s_a$ is $pq^\omega$ where $\omega \to \infty$ represents the number of times the shortest cycle is executed. The cost of the optimal path is calculated as $w(p) + w(q)^\omega$ where $w(p)$ or $w(q)$ is the sum of the weights of the transitions on $p$ or $q$. In the implementation, we use $\omega = 5$ to estimate the path cost.
- Finally, we select the optimal path on $G_{\mathcal{B}}$ as the least-cost path to any accepting state of $G_{\mathcal{B}}$.

**5.** For task execution, on the searched optimal path $pq^\omega$, we use a goal-conditioned agent to subsequently achieve the goals along the path $p$ and reach the goals on $q$ iteratively in a loop to satisfy $\varphi$. For any transition $(q_i, \psi, q_j)$ on a searched path, the agent employs its goal-conditioned policy $\widetilde{\pi}(\cdot|s, \psi)$ to reach the targeted goal region $\psi$ (Fig. 4(d),(e)).

**Justification.** Recall that our objective of learning a LTL specification satisfying policy (Equation 3) is finding $\pi^* = \arg\max_\pi \mathbb{E}_{\tau \sim \pi(\cdot|\cdot, \varphi)} [\mathbb{1}[\tau \models \varphi]]$, which generates the maximum number of LTL-satisfying runs for a given LTL property $\varphi$. Our formalization of goal-condition RL uses a sparse

binary reward function (Equation 1) - a reward of 1 is provided only when the specified goal is successfully achieved by the end of a training episode. In this setting, when the discounted factor $\gamma \to 1$, the weight $w$ of a transition on $G_{\mathcal{B}}$, which is determined by the learned value function $\mathcal{V}$ (e.g. $w = -\log \mathcal{V}(\cdot)$), is inversely proportional to the probability of reaching the goal region represented by the target node from that represented by the source node. As such, based on the capability of the goal-conditioned agent (i.e. the learned value function $\mathcal{V}$), the task planning algorithm seeks the optimal path as the one the agent is most likely to succeed.

### 3.3 Handling Avoidance

As depicted in Fig. 4(d), our method uses the goal-conditioned agent policy $\widetilde{\pi}$ to fulfill a reach-avoid subtask defined as $\widetilde{\pi}(\cdot|s, \wedge_j g_j, \wedge_k \neg g_k)$ - from a state $s$ an agent is required to avoid several regions $g_k$ ($k \geq 1$) before reaching the target region $\wedge_j g_j$ in the goal space. Our method only performs avoidance in situations where there is a high likelihood of colliding with any $g_k$. The likelihood of a collision occurring with $g_k$ from a current state $s$ can be assessed by evaluating the value function $V(s, g_k)$ of the goal-conditioned agent. If $V(s, g_k)$ exceeds a threshold $\sigma$, it indicates a high likelihood of collision. We formally define the reach-avoid policy $\widetilde{\pi}$ as follows:

The strategy is to find the most dangerous zone $g_k$ to avoid. A predicted value $V(s, g_k)$ below the threshold $\sigma$ (e.g., 0.85) signifies that $g_k$ is not considered dangerous and does not need to be avoided. We then take a goal-reaching action $\arg\max_a \min_j Q^\pi(s, g_j, a)$ Tasse et al. [2020] for reaching the overlapped goal space covered by $\wedge_j g_j$. Otherwise, we select a *safe action* $s = \arg\min_a Q(s, g_k, a)$ that moves the agent away from the dangerous zone $g_k$ and identify a *dangerous action* $d = \arg\max_a Q(s, g_k, a)$ that moves the agent directly to the dangerous zone $g_k$. We then select a "blocked" goal-reaching action $t = \arg\max_{a \neq d} \min_j Q^\pi(s, g_j, a)$ to eliminate the impact from the *dangerous action* $d$. For continuous environments, we generate the combined action $s + t$ that accounts for both safety and reachability. For discrete environments, we simply return $t$ to block the effect of unsafe actions.

$$\widetilde{\pi}(\cdot|s, \bigwedge_j g_j, \bigwedge_k \neg g_k) \equiv$$
$$\quad \textbf{let } k = \arg\max_k V(s, g_k) \textbf{ in}$$
$$\quad \textbf{if } \text{if } V(s, g_k) < \sigma$$
$$\quad \textbf{then } \arg\max_a \min_j Q^\pi(s, g_j, a)$$
$$\quad \textbf{else}$$
$$\qquad \textbf{let } d = \arg\max_a Q(s, g_k, a)$$
$$\qquad \textbf{let } s = \arg\min_a Q(s, g_k, a)$$
$$\qquad \textbf{let } t = \arg\max_{a \neq d} \min_j Q^\pi(s, g_j, a)$$
$$\qquad s + t$$

Figure 5: Handling reach-avoid subtasks by goal-conditioned agents in GCRL-LTL.

## 4 Experiments

We implemented our algorithm in a tool called GCRL-LTL[2] that extends goal-condition RL to task-oriented RL where tasks are specified by LTL objectives.

We evaluate GCRL-LTL in the **ZoneEnv** environment shown in Fig. 1 and the **Ant-16rooms** environments depicted in Fig. 3. We also include a 7×7 discrete grid-based environment **LetterWorld** from Andreas et al. [2017]. In this environment, 12 unique letters occupy 24 positions out of 49 squares, where each letter is presented twice. This denotes the agent can satisfy an atomic proposition by reaching any of the two. The initial positions of the robots and the zones or letters are random in every episode. More information about these environments is provided in Appendix G.1.

**Primitives.** As shown in Fig. 5, our agent policy requires enumerating the action space to perform an avoidance maneuver when a collision is imminent. To support efficient enumeration over continuous action space, we pretrain four neural primitive skills that can move the Point robot in ZoneEnv and Mujoco ant in Ant-16rooms along the four cardinal directions *UP, DOWN, LEFT, RIGHT*. Each of the skills is trained for 0.5 million steps. More details for primitive training are given in Appendix G.2. To ensure a fair comparison, both our learning algorithm and the baselines get access to the neural primitives, which enable a discrete action space for both.

---

[2]GCRL-LTL is available at `https://github.com/RU-Automated-Reasoning-Group/GCRL-LTL`

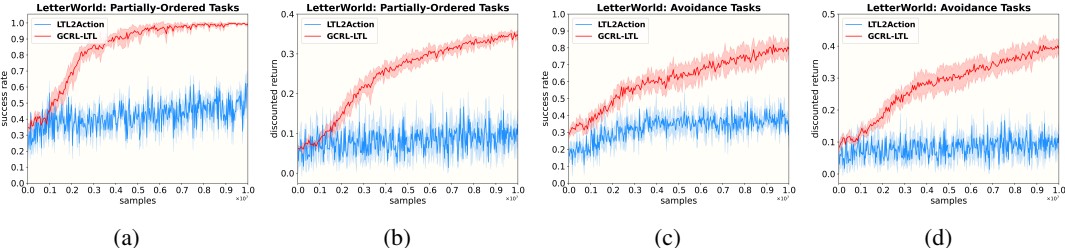

(a)        (b)        (c)        (d)

Figure 6: (a) and (b) shows success rates and discounted rewards of `LetterWorld` on partially-ordered tasks, respectively. (c) and (d) shows success rates and discounted rewards on avoidance tasks, respectively. We report the average results on 5 random seeds, where each data point is acquired by collecting results from 100 episodes.

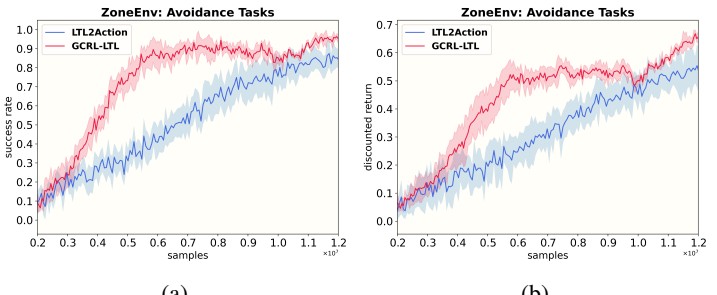

(a)             (b)

Figure 7: Comparisons of (a) success rate and (b) discounted rewards in `ZoneEnv` on avoidance tasks. We report the average results on 5 random seeds, where each data point is acquired by collecting results from 50 episodes. The initial 2 million steps are omitted due to the training of the dynamic primitives.

We extend Proximal Policy Optimization (PPO) (Schulman et al. [2017]) to a goal-conditioned RL algorithm to train the goal-conditioned agents for `LetterWorld` and `ZoneEnv`. We use an extension of the GCSL Ghosh et al. [2021] algorithm to train a goal-conditioned agent for `Ant-16rooms`. More details about the training algorithms are provided in Appendix B.

## 4.1 Evaluation in Multi-Task Settings

**Baseline.** We compared GCRL-LTL with **LTL2Action** ( Vaezipoor et al. [2021]). This baseline exemplifies the state-of-the-art learning algorithms for LTL-satisfying policies that use advanced neural architectures to encode LTL task formulas for policy decision-making. In contrast, GCRL-LTL employs weighted graph search to decompose LTL tasks into basic reach-avoid tasks that can be accomplished by goal-conditioned agents.

We consider the following LTL task spaces to evaluate GCRL-LTL and the baseline.

**Partially-Ordered Tasks**. Such a task consists of multiple streams of sub-tasks. A sub-task contains a sequence of goals that must be satisfied in a specified order. For instance, a possible task can be $\mathbf{F}(p \wedge \mathbf{F}(q \wedge \mathbf{F}r)) \wedge \mathbf{F}(s \wedge \mathbf{F}q)$. This task can be described as *"satisfy p, q, r in that order; and satisfy s, q in that order"*. One valid solution is to reach *s, p, q, r* in order to satisfy all sub-tasks.

**Avoidance Tasks**. Similar to partially ordered tasks, a sequence of propositions must be satisfied in order, while some propositions must be avoided. Visiting a prohibited position leads to failure. An example avoidance task is $\neg j\mathbf{U}(w \wedge (\neg y\mathbf{U}r))$ (Fig. 2), which can be interpreted as a two-stage task: *"reach w while avoiding j in the first stage, then reach r while avoiding y in the second stage"*.

During training, LTL2Action has access to a task sampler that produces a random LTL task from the large set of possible task spaces described above for each training episode. GCRL-LTL does not use the task sampler and is only trained to reach random goals. In evaluation, tasks are randomly sampled in each episode for both agents. We choose $\sigma = 0.85$ in Fig. 5 and perform ablation study on this value in Appendix G.3.

Figure 6 shows the evaluation results across training iterations on both partially ordered and avoidance tasks in `LetterWorld`. Fig. 7 demonstrates the results for avoidance tasks in `ZoneEnv`. GCRL-LTL outperforms LTL2Action by a large margin despite having no access to the task sampler during training. Furthermore, we assess the generalizability of GCRL-LTL and the baseline by testing

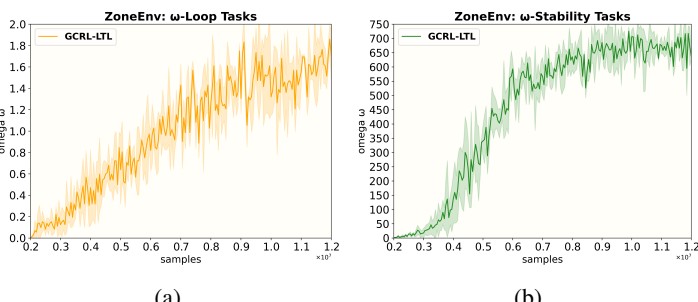

(a)  (b)

Figure 8: Figures (a) and (b) depict the performance of trained goal-conditioned agents in satisfying the $\omega$-loop and $\omega$-stability tasks in the ZoneEnv environment. Each data point represents the average performance obtained from evaluation on 20 episodes, and the results are averaged over 5 random seeds.

| Environment | Tasks | Timeout | Ours | LTL2Action |
|---|---|---|---|---|
| LetterWorld | Partially-Ordered | 75 | **0.996(0.423)** | 0.661(0.129) |
| | Partially-Ordered + Depth $\Uparrow$ | 225 | **0.986(0.259)** | 0.477(0.033) |
| | Partially-Ordered + Conjuncts $\Uparrow$ | 225 | **0.992(0.410)** | 0.567(0.049) |
| ZoneEnv | Avoidance | 1000 | **0.958(0.632)** | 0.934(0.601) |
| | $\mathbf{F}(a \wedge \mathbf{X}(\neg b \mathbf{U} c)) \wedge \mathbf{G}(\neg d)$ | 1000 | **0.928(0.543)** | 0.342(0.304) |
| | $\mathbf{F}(a \wedge \mathbf{F}(b \wedge \mathbf{F}(c \wedge \mathbf{F} d)))$ | 2000 | **0.926(0.220)** | 0.060(0.020) |

Table 1: Performance of trained agents on original and out-of-distributions tasks. We report the *success rate* and *discounted return*(in parentheses). Results are averaged over 1000 episodes.

them on more complex tasks that are out-of-distribution to the task sampler used in baseline training. Specifically, for partially ordered tasks in LetterWorld, we increase the maximum depth from 5 to 15, and the number of conjunctions from 4 to 12. Similarly, we extend the avoidance tasks in ZoneEnv to a more difficult setting where the agent needs to avoid one extra zone in the second stage ($\mathbf{F}(a \wedge \mathbf{X}(\neg b \mathbf{U} c)) \wedge \mathbf{G}(\neg d)$). Additionally, we test if agents trained in ZoneEnv can efficiently solve tasks in which a long sequence of consecutive *zones* must be reached ($\mathbf{F}(a \wedge \mathbf{F}(b \wedge \mathbf{F}(c \wedge \mathbf{F} d)))$). The experimental results, as detailed in Table 1, indicate that GCRL-LTL significantly outperforms the baseline in its ability to generalize to unseen tasks.

$\omega$-**regular LTL Tasks.** We conducted experiments to evaluate the performance of GCRL-LTL in satisfying $\omega$-regular LTL objectives. $\omega$-**Stability Tasks**: such a task requires the agent to satisfy a task proposition by reaching a goal region and remaining in the goal region infinitely. $\omega$-**Loop Tasks:** such a task in the form $\mathbf{GF}(a \wedge \mathbf{X} \mathbf{F} b) \wedge \mathbf{G}(\neg c)$ requires the agent to continuously cycle between two specific zones, $a$ and $b$, while always avoiding another zone, $c$. For example, in the task $\mathbf{GF}(r \wedge \mathbf{X} \mathbf{F} y) \wedge \mathbf{G}(\neg w)$, the agent is expected to traverse between the *Red* and *Yellow* zones indefinitely, while never visiting any *White* zone. We evaluate the satisfiability of $\omega$-loop and $\omega$-stability tasks in the ZoneEnv environment during the goal-conditioned agent training process. The results are presented in Figure 8. The *x*-axis

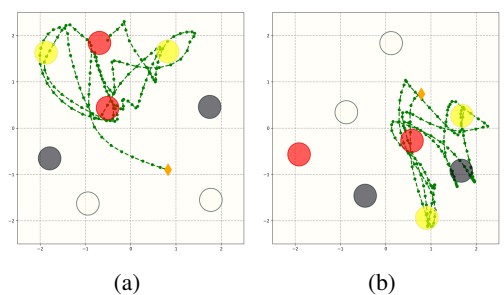

(a)  (b)

Figure 9: Figures (a) and (b) illustrate agent trajectories that solve the $\omega$-loop task $\mathbf{GF}(r \wedge \mathbf{X} \mathbf{F} y) \wedge \mathbf{G}(\neg w)$ in the ZoneEnv environment. The initial positions of the robot are represented by orange diamonds ♦.

represents the number of environment interaction steps, while the *y*-axis represents the number of $\omega$ occurrences. In the $\omega$-loop tasks it represents the number of rounds completed by the agent in looping between specific zones, and in the $\omega$-stability tasks it represents the number of transitions the agent successfully remains in the desired zone, within the first 1500 timesteps (higher values indicate better performance). The results confirm that GCRL-LTL can handle $\omega$-regular LTL specifications even though the underlying goal-conditioned agents have never seen such specifications during their training time. Figures 9a and 9b illustrate agent trajectories that solve the above $\omega$-LTL task. In both

trajectories, the robot quickly reaches the *Red* zone and subsequently oscillates between the *Red* and *Yellow* zones. More examples can be found in Appendix E.

## 4.2 Comparison with Compositional RL

We also compare our approach with DiRL ( Jothimurugan et al. [2021]), which is a state-of-the-art compositional RL algorithm for LTL-satisfying policies. DiRL simultaneously learns a low-level policy for each sub-goal transition and a high-level policy to plan over the sub-goal space. However, it is restricted to a small fragment of LTL, cannot handle $\omega$-regular LTL properties (e.g. Fig. 1), and cannot be applied to multi-task RL because the low-level policies are specific to a single environment setting and do not generalize across different environments. We used 8 LTL task specifications. These tasks have increasing levels of difficulty as they require the agent to sequentially reach a growing number of sub-goals. An example instruction is given in Fig. 3. DiRL has to train a separate agent for each of the LTL specifications. In contrast, GCRL-LTL trains a single goal-conditioned agent and evaluates this agent over all 8 LTL specifications. Our agent achieves a success rate of approximately 90% in fulfilling all specifications after being trained with 3e6 environment steps, whereas the DiRL method has to exercise 3e6 steps for each of the specifications to match the success rate of GCRL-LTL. More comprehensive evaluation results are given in Appendix F.

## 5 Related Work

There exists a large amount of previous work in using linear temporal logic (LTL) for specifying or shaping the reward functions for reinforcement learning (RL) Aksaray et al. [2016], Littman et al. [2017], Sadigh et al. [2014], Hasanbeig et al. [2018], Cai et al. [2021], Hahn et al. [2018], Camacho et al. [2019], Hasanbeig et al. [2020], Jothimurugan et al. [2020], Icarte et al. [2018, 2022]. These methods primarily concentrate on learning a single, specific task defined by an LTL objective. In contrast, our approach involves learning task-oriented policies akin to goal-conditioned RL agents within a multitasking framework. In this setup, a new *arbitrarily complex* LTL task is sampled as a goal for each episode.

Past research efforts have also aimed to empower RL agents to handle previously unseen multi-task instructions expressed in LTL. One common solution is decomposing complex LTL tasks into smaller, independent subtasks to simplify the learning process. For example, Araki et al. [2021] develops a hierarchical options framework that learns various neural networks each specialized for one subtask, and optimally composes these learned options via value iteration to solve temporal logic specifications. However, subtask execution does not consider what the agent must do afterward and hence might perform sub-optimally in solving the full task. León et al. [2022] progresses towards completion of an LTL task by greedily identifying subtasks of the original LTL instruction that remain to be addressed, which may result in myopic behaviors. Recent works show that the agents' neural architecture to embed LTL specifications is the key to improving the performance of RL agents in unseen environments. Kuo et al. [2020] presents a novel network architecture framework to compose neural networks one for each LTL operator and environment object to mirror the formula structure. In a similar vein, Vaezipoor et al. [2021] uses neural encodings to interpret full LTL specifications to guide an RL agent to learn task-conditioned policies in multitask settings. Voloshin et al. [2023] developed eventual discounting that exploits the fact that optimally satisfying the LTL specification does not depend on the length of time it takes to reach accepting states (e.g., "eventually always reach the goal") to find LTL-satisfying policies with the highest achievable probability. We discuss the other related work in a more general context in Appendix C.

## 6 Conclusion

This paper presents GCRL-LTL to learn task-oriented policies where tasks are specified by Linear Temporal Logic (LTL) instructions. Unlike existing methods that sample a vast set of LTL instructions during training, our approach enables simple goal-conditioned RL agents to follow arbitrary and complex LTL specifications without the need for additional training over the LTL task space. GCRL-LTL is capable of handling diverse and intricate LTL specifications including $\omega$-LTL objectives over infinite time horizons. Through extensive experiments, we demonstrated the effectiveness of GCRL-LTL to seamlessly satisfy complex temporal logic task specifications in a zero-shot manner.

## Acknowledgements

We thank the anonymous reviewers for their comments and suggestions. This work was supported by NSF Award #CCF-2124155 and NSF Award #CCF-2007799.

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

**Appendix**

# Table of Contents

# A GCRL-LTL Limitations

The main limitation of our method is that we assume the atomic propositions in LTL properties can only be goals within the goal space of the underlying goal-conditioned policy e.g. colored zones in ZoneEnv navigation. We do not allow other sources of atomic propositions e.g. external environment signals that are out of the agent's control. For example, our current algorithm does not apply when the agent needs to pursue different tasks based on an external signal.

A potential solution to the aforementioned limitation is using a task monitor, which acts as an external memory, to maintain a record of completed sub-goals and past external environment signals. During task execution, when receiving a new environmental signal, our task planning algorithm can dynamically revise the high-level path for the remaining sub-goals that the goal-conditioned agent needs to achieve. We leave it for future work.

The other limitations of GCRL-LTL include (1) the overapproximation of the true optimal probability by our task-planning algorithm for $\omega$-regular LTL properties by bounding $\omega$ to a finite number in Sec. 3.2; (2) dynamic handling of avoidance during test time instead of during planning - our task-planning algorithm does not consider the goal-conditioned agent's ability to stay safe before reaching a sub-goal; (3) the lack of an alarm system for unsatisfiable task specifications. We experimented with unsatisfiable task specifications. For properties such as $\mathbf{F}(a \wedge b)$, when the goal regions $a$ and $b$ are close but not overlapping the agent's behavior mirrors that of $\mathbf{GF}\, a \wedge\, \mathbf{GF}\, b$, oscillating between $a$ and $b$. However, if these goal regions are far apart, our agent does not exhibit good-looking behavior.

# B Pseudo Code for Goal-Conditioned Reinforcement Learning

## B.1 Pseudo Code for the Goal-Conditioned PPO Algorithm

Algorithm 1 illustrates the pseudocode for our goal-conditioned proximal policy optimization (PPO) algorithm. This algorithm builds upon the standard PPO algorithm (Schulman et al. [2017]). Notable variations in the training process include the following: (1) when gathering experiences from the environment, it is crucial to synchronize the environment, goal representation, and the reward function; (2) an extended version of the value function $V_\phi(s, g)$ and policy $\pi_\theta(s, g)$ is necessary, which take both an environment state $s$ and one of the goal representations $g$ as inputs.

Including the goal as an input in goal-conditioned reinforcement learning algorithms enables the agent to explicitly incorporate the desired objectives into its decision-making process and to learn a policy that is explicitly aware of the desired goals. As such, the policy provides a mechanism for the agent to condition its behavior on the desired outcome, allowing it to generate actions that are tailored to the specific goal at hand. When the goal is included as an input to the value function, it helps the agent estimate the value of being in a particular state while aiming for a specific goal. This assists in determining the quality or potential of a state.

As discussed in the paper, we leverage goal-conditioned RL agents to interact with low-level environments to learn how to achieve basic goals without considering higher-level tasks. To solve an unseen LTL instruction, we engage in high-level reasoning over the difficulty of achieving the specified goals in the goal space to optimally achieve the sequences of tasks specified in the LTL objective. To this end, we learn a function $\mathcal{V} : S \times \mathcal{G} \times \mathcal{G} \to \mathbb{R}$ to estimate the capability of taking the goal-conditioned policy $\pi$ to go from one goal to another goal on average. We regress $\mathcal{V}$ towards the value function of the a goal-conditioned agent by minimizing:

$$\min_{\mathcal{V}} \left(V(s_t, g_k) - \mathcal{V}(s_j, g_t, g_k)\right)^2 \quad \text{where } g_k \in L(s_k) \wedge g_t \in L(s_t)$$

with $\tau \sim B$ (a replay buffer), $t \sim \{0 \ldots t_{max}\}$, $s_t, a_t, s_{t+1} \sim \tau$, $j \sim \{0, t\}$, $k \sim \{t + 1 \ldots t_{max}\}$, and $L$ being given as the goal labeling function. In Algorithm 1, we train $\mathcal{V}$ together with the goal-conditioned PPO algorithm.

## B.2 Pseudo Code for the Goal-Conditioned Supervised Learning Algorithm

Algorithm 2 extends goal-conditioned supervised learning (GCSL) ( Ghosh et al. [2021]), following its iterative procedure of sampling trajectories, relabeling them, and training a policy until convergence. It trains an agent to achieve specific goals in a given environment. The primary enhancement

---

**Algorithm 1** Goal-Conditioned Proximal Policy Optimization Algorithm

---

**Require:** Initial policy parameters $\theta_0$, initial value function parameters $\phi_0$, initial Q function parameters $\beta$, initial goal value function parameters $\omega$, goal-labeling function $L : S \to \mathcal{G}$ from state space $S$ to goal space $\mathcal{G}$, discount factor $\gamma$, clip range $\epsilon$

**Ensure:** Goal-conditioned policy $\pi_\theta(s, g)$

  **for** $k = 0, 1, 2, \ldots, N$ **do**

    $\mathcal{D}_k = \{\}$

    **for** $m = 0, 1, 2, \ldots, M$ **do**                                   $\triangleright$ Collect goal-conditioned trajectories

      Sample $g_m \sim \mathcal{G}$, $s_0 \sim S$

      $\mathcal{D}_k^m = \{\tau_m = (\ldots, (s_i, g_m, a_i, r_i), \ldots)\}$

      $\mathcal{D}_k = \mathcal{D}_k \cup \mathcal{D}_k^m$         $\triangleright$ Acquire trajectories by running policy $\pi_k(s, g_m) = \pi(\theta_k)$

    **end for**

    Compute rewards-to-go $\hat{R}_t$

    Compute advantage function $\hat{A}_t$ based on the current value function $V_{\phi_k}$

                                                        $\triangleright$ Update the policy $\pi_\theta$

$$\theta_{k+1} = \arg\max_\theta \frac{1}{|\mathcal{D}_k|T} \sum_{\tau \in \mathcal{D}_k} \sum_{t=0}^{T} \min\left(\frac{\pi_\theta(a_t|s_t, \tau_g)}{\pi_{\theta_k}(a_t|s_t, \tau_g)} A^{\pi_{\theta_k}}(s_t, \tau_g, a_t), \ \ f(\epsilon, A^{\pi_{\theta_k}}(s_t, \tau_g, a_t))\right),$$

where $\tau_g$ is the goal of rollout $\tau$ and $f(\epsilon, A) = \begin{cases} (1+\epsilon)A, & A \geq 0 \\ (1-\epsilon)A, & A < 0 \end{cases}$

                                                        $\triangleright$ Update the $Q$ function $Q_\beta$

$$\beta_{k+1} = \arg\min_\beta \frac{1}{|\mathcal{D}_k|T} \sum_{\tau \in \mathcal{D}_k} \sum_{t=0}^{T} (Q_\beta(s_t, \tau_g, a_t) - (r_t + \gamma V_{\phi_k}(s_{t+1}, \tau_g)))^2$$

                                                $\triangleright$ Update the goal value function $\mathcal{V}_\omega$

$$\omega_{k+1} = \arg\min_\omega \frac{1}{|\mathcal{D}_k|T} \sum_{\tau \in \mathcal{D}_k} \sum_{t=0}^{T} \sum_{j=0}^{t-1} \sum_{k=t+1}^{T} (V_{\phi_k}(s_t, g_k) - \mathcal{V}_\omega(s_j, g_t, g_k))^2$$

where $g_k = L(s_k)$, $g_t = L(s_t)$, and $s_0$ is the initial state of $\tau$

                                                    $\triangleright$ Update the value function $V_\phi$

$$\phi_{k+1} = \arg\min_\phi \frac{1}{|\mathcal{D}_k|T} \sum_{\tau \in \mathcal{D}_k} \sum_{t=0}^{T} (V_\phi(s_t, \tau_g) - \hat{R}_t)^2$$

  **end for**

---

involves organizing past goals into a graph structure $G$ based on their reachability to improve agent exploration. This graph represents connections between goals, with nodes representing past goals and edges indicating that one goal can be reached from another. Our algorithm explicitly uses the goal value function $\mathcal{V}(s_0, g_1, g_2)$ (which is learned together with the policy) to measure the capability of reaching a goal node $g_2$ from a goal node $g_1$ from the viewpoint at an initial state $s_0$.

Assume $L$ is a goal-labeling function from state space $S$ to goal space $\mathcal{G}$. Algorithm 2 works as follows. (1) Its samples a goal from a goal space $\mathcal{G}$ and an initial state from a state space $S$. (2) It collects a trajectory $\tau = (s_0, a_0, s_1, \ldots, s_t)$ to a goal node $g'$ on $G$ that is the closest node to $g$ (distances between goal nodes are measured by $\mathcal{V}$) until either $L(s_t) = g'$ or $t$ exceeds the maximum horizon. During this step, at each current state $s$, it finds the shortest past $\rho$ from $L(s)$ to $g'$, where $\rho = L(s), g_{i+1}, g_{i+2}, \ldots, g'$ and then calling the policy $\pi(\cdot|s, g_{i+1})$ to reach the goal $g_{i+1}$. (3) It executes the current policy $\pi$ for $T - t$ steps in the environment to collect a trajectory $\tau'$ to reach the sampled goal $g$. (4) It concatenates $\tau$ and $\tau'$ and relabels the trajectory to add new expert tuples $(s_t, g = s_{t+h}, a_t, \hat{R}^t)$ for $t, h > 0, t + h \leq T$ to the training dataset. (5) It performs supervised learning on the entire dataset to update the goal value function $\mathcal{V}$ and the policy $\pi$ via maximum likelihood.

## C   Additional Related Work

### C.1   Goal-conditioned Reinforcement Learning

Goal-conditioned Reinforcement Learning(GCRL) is a popular topic that has been studied for a long time in many previous works. To characterize Goal-Augmented MDP, Schaul et al. [2015] provides a straightforward method that extends the standard value function to a goal-conditioned value function,

---
**Algorithm 2** GCSL with organizing past goals into a graph structure $G$

---
**Require:** Initial goal value function $\mathcal{V}(s_0, g_1, g_2)$, initial policy $\pi(\cdot|s, g)$
**Require:** Maximal horizon $T$, discounted factor $\gamma$, (unknown) environment transition relation $\mathcal{P}$,
   goal-labeling function $L : S \to \mathcal{G}$ from state space $S$ to goal space $\mathcal{G}$
**Require:** Goal graph $G(\mathcal{G}, E)$ with nodes in $\mathcal{G}$, edges $E$, and edge weight $W_e$ for $e \in E$
   **for** $k = 1, 2, 3...$ **do**
      Sample $g \sim \mathcal{G}$, $s_0 \sim S$
      $g' = \arg\min_{g' \in V} \mathcal{V}_k(s_0, g', g)$
      Log a trajectory $\tau = (s_0, a_0, s_1, a_1, \ldots, s_t)$ until $L(s_t) = g'$ or $t \geq T$ with $s_{i+1} \sim \mathcal{P}(s_i, a_i)$,
        $a_i \sim \pi_k(s_i, g_{i+1})$ where $g_{i+1}$ is on the shortest path $\rho$ from $L(s_i)$ to $g'$ in $G$ w.r.t edge weight
        $W_{g_1 \to g_2} = \mathcal{V}_k(s_0, g_1, g_2)$ for any nodes $g_1, g_2 \in \mathcal{G}$, and $\rho = L(s_i), g_{i+1}, g_{i+2}, \ldots, g'$.
      Log trajectory $\tau' = (s_t, a_t, \ldots, S_T)$ where $s_{i+1} \sim \mathcal{P}(s_i, a_{i+1})$, $a_{i+1} \sim \pi_k(s_i, g)$
      $\tau = \tau \ @ \ \tau'$                                 $\triangleright$ Concatenate $\tau$ and $\tau'$
                                                       $\triangleright$ Update goal graph $G$
      For each state $s_i$ on $\tau$, add $L(s_i)$ to vertices $V$, add $(L(s_i), L(s_{i+1}))$ to edges $E$ of $G$,
      $D_\tau = \{(s_t, g = s_{t+h}, a_t, \hat{R}_t) : t, h > 0, t + h \leq T\}$ with $\hat{R}_t = \sum_{k=0}^{h} \gamma^k r(s_{t+k}, g, a_{t+k})$
      $D_{k+1} = D_\tau \cup D_k$
      $\mathcal{V}_{k+1} = \arg\min_{\mathcal{V}} \mathbb{E}_{(s_t, g, \_, \hat{R}_t) \in \mathcal{D}_{k+1}} (\mathcal{V}(s_0, L(s_t), g) - \hat{R}_t)^2$    $\triangleright$ Update the goal value function
      $\pi_{k+1} = \arg\max_{\pi} \mathbb{E}_{(s_t, g, a_t, \_) \in \mathcal{D}_{k+1}} [\log_\pi(a_t \mid s_t, g)]$             $\triangleright$ Update the policy
   **end for**

---

called universal value function approximation (UVFA). UVFA serves as a basis in many following research works. How to define an appropriate reward is one of the basic challenges in GCRL. Some work reshapes the reward with the distance between the achieved goal and the desired goal but without any knowledge of the environment, i.e. Trott et al. [2019]. However, the local optimal policy it produces cannot finish many of the tasks in different environments. To solve long-horizon tasks, many planning methods on the abstract level above the robot control level are proposed. Nasiriany et al. [2019] introduced a goal-conditioned value function to measure the reachability of the next sub-goal given the previous sub-goal by analyzing how close the goal-conditioned policy gets to the goal after a specific number of time steps. Eysenbach et al. [2019] broke apart the task of reaching a distant goal into a sequence of way-points selected by learning a distributional Q-value function which indicates how many steps the current state and goal are away from one another. For the data as images, Chane-Sane et al. [2021] imagines possible sub-goals in a self-supervised fashion and uses them to facilitate training. By embedding goals into a latent space that captures some notion of temporal distance between goals, Zhang et al. [2021] learns a set of latent landmarks scattered across the goal space to enable scalable planning.

More generally, motivated by the fact that human learns from failure experience, Andrychowicz et al. [2018] provided a famous approach called Hindsight Experience Replay (HER), which can relabel the desired goals in the replay buffer with any reached goal within the same trajectory. Moreover, HER mitigates the sparse reward issue. In addition, many works try to extend the idea of HER. For instance, Rauber et al. [2019] demonstrates how hindsight can be introduced to policy gradient methods and generalize this idea to a successful class of reinforcement learning algorithms; Drawn intuition from the observation that any trajectory is a successful trial for reaching its final state, Ghosh et al. [2021] proposed a self-imitation algorithm named goal-conditioned supervised learning (GCSL). To enhance the diversity of relabeled goals, Zhu et al. [2021] develop a model-based reinforcement learning approach with a new relabeling strategy that relabels the goals by looking into the future with a learned dynamics model.

### C.2 Reinforcement Learning Meets Logic and Specifications

In recent years, instructing agents using LTL formulas has been studied in a wide range of research works (Aksaray et al. [2016], Littman et al. [2017], Sadigh et al. [2014], Hasanbeig et al. [2018], Cai et al. [2021], Hahn et al. [2018], Camacho et al. [2019]). Among these approaches, retrieving information from and handling Q values in reinforcement learning is the mainstream implementation.

However, these Q-learning based approaches are not suitable for complex environment with continuous action space, until it is extended by Hasanbeig et al. [2020]. The above methods typically analyze a given LTL specification and then formulate a (usually sparse) reward function correspondingly. Then it could be used by any RL algorithm to learn a policy. Specifically, Jothimurugan et al. [2020] designed and implemented a domain-specific language for specifications called SPECRL. Users may decompose a complex task with sequences, disjunctions and (or) conjunctions of simpler sub-tasks, and can define the safety properties. Based on the above information, they construct a finite state machine named *task monitor* that generates shaped rewards. Icarte et al. [2018] build a model based on automaton called *reward machines* (RM). The RMs are used to represent and decompose complex high-level LTL specifications, so that a curated RL algorithm called QRM may be applied to exploit the structure of the RMs. In their following work of Icarte et al. [2022], they propose a hierarchical RL algorithm called HRM that acquires policies using RMs. Besides research works focusing on classic single-agent RL environments, there are a few recent works on RL with logic specifications in multi-agent RL environments (Hammond et al. [2021]). Still, experiments show training RL agents using classic RL algorithms without considering the formulation and semantics of LTL may lead to sub-optimal myopic policies (Vaezipoor et al. [2021], Icarte et al. [2022]). To this end, policy sketches may be used in achieving the LTL formula, where a sequence of sub-tasks is relatively easier to solve than a complex task as a whole. Andreas et al. [2017] show that such sketches are useful when solving long-horizon tasks. Sun et al. [2020] and Zhao et al. [2021] show that injecting semantics to a programmatic agent can speed up learning for complex long-horizon tasks.

# D    Demonstrations of the LTL Tasks in Sec. 4

In this section, we provide a detailed explanation of the general format for each Linear Temporal Logic (LTL) task presented in the paper. Additionally, we present specific examples of trajectories that illustrate how our method addresses a diverse range of challenging tasks across various environments. These trajectories serve as concrete demonstrations of our method's effectiveness in tackling these tasks.

## D.1    Avoidance Tasks

The avoidance task follows a general format represented as $\neg a\mathbf{U}(b \wedge (\neg c\mathbf{U}d))$, where $a, b, c, d$ correspond to propositions or goals relevant to task completion in our context. In Figures 10a and 10b, we showcase two trajectories generated by our trained agent to solve a specific avoidance task: $\neg y\mathbf{U}(j \wedge (\neg w\mathbf{U}r))$. This task can be intuitively understood as a two-stage process: the agent needs to reach the "Jetblack" zone while avoiding the "Yellow" zone in the first stage, and subsequently reach the "Red" zone while avoiding the "White" zone in the second stage. Figures 10a and 10b serve as clear demonstrations of the agent's ability to reach the desired zones while never entering the prohibited zones.

## D.2    Avoiding More Tasks

The avoiding more task follows a general format represented as $\mathbf{F}(a \wedge \mathbf{X}(\neg b\mathbf{U}c)) \wedge \mathbf{G}(\neg d)$, where $a, b, c, d$ are task propositions or goals in our context. Figures 10c and 10d illustrate trajectories generated by our trained agent to solve an avoiding more task: $\mathbf{F}(j \wedge \mathbf{X}(\neg y\mathbf{U}r)) \wedge \mathbf{G}(\neg w)$. Similar to avoidance tasks, avoiding more tasks requires the agent to traverse a sequence of zones in two stages while avoiding specific zones. The distinction is that avoiding more tasks require the agent to avoid one additional zone in the second stage, making it more challenging. Informally, the example can be interpreted as follows: in the first stage, the agent needs to reach the "Jetblack" zone while avoiding the "White" zone; in the second stage, the agent must reach the "Red" zone while avoiding the "White" and "Yellow" zones. The trajectories showcased in Figures 10c and 10d demonstrate the ability of our trained agent to effectively navigate through the required zones, avoiding the designated zones as specified by the avoiding more task.

From Figures 10a and 10c, we can observe how the presence of additional zones to avoid affects the trajectories, given identical initial states and sequences of goals (*Jetblack* and *Red*). Specifically, in the second stage of the avoidance task $\neg y\mathbf{U}(j \wedge (\neg w\mathbf{U}r))$, the agent is expected to reach the *Red* zone while visiting a *Yellow* zone is not prohibited. However, in the second stage of the avoid more task $\mathbf{F}(j \wedge \mathbf{X}(\neg y\mathbf{U}r)) \wedge \mathbf{G}(\neg w)$, it is not allowed to visit any *Yellow* zone. In order to avoid the *Yellow*

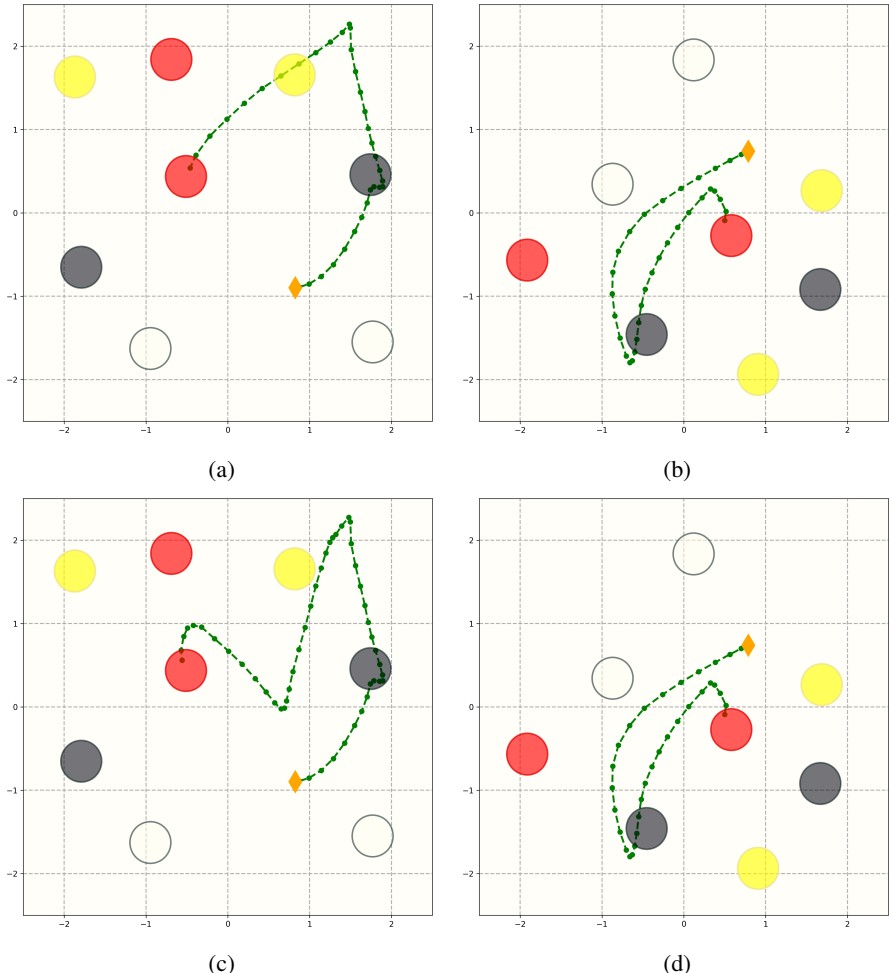

Figure 10: Figures (a) and (b) depict trajectories that solve the avoidance task: $\neg y\mathbf{U}(j \wedge (\neg w\mathbf{U}r))$. On the other hand, Figures (c) and (d) showcase trajectories that solve the avoiding more task: $\mathbf{F}(j \wedge \mathbf{X}(\neg y\mathbf{U}r)) \wedge \mathbf{G}(\neg w)$. The initial positions of the robot are represented by orange diamonds ◆. The different colors of the zones, namely *Red, Yellow, White*, and *Jetblack*, correspond to the task propositions *r, y, w, j*, respectively. It is important to note that the initial positions of the robot and the zones are randomized. The initial states of Figures (a) and (c) are the same, while the initial states of Figures (b) and (d) are identical.

zones in the second stage of the avoid more task, the agent takes additional steps (as shown in Figure 10c) to bypass a *Yellow* zone and eventually reach a *Red* zone. Figures 10b and 10d demonstrate that identical trajectories can solve different LTL tasks. This occurs when the extra zones to avoid are located far away from the robot throughout the entire execution, rendering no impact on the trajectories.

### D.3 Goal Chaining Tasks.

The goal chaining task follows a general format expressed as $\mathbf{F}(a \wedge \mathbf{F}(b \wedge \mathbf{F}(c \wedge \mathbf{F}d)))$, where $a, b, c, d$ represent propositions or goals that are relevant to task completion in our context. Figures 11a and 11b display the trajectories for solving goal chaining tasks, specifically $\mathbf{F}(j \wedge \mathbf{F}(w \wedge \mathbf{F}(r \wedge \mathbf{F}y)))$. In this particular LTL task, the agent is required to navigate through zones with the colors *Jetblack, White, Red,* and *Yellow* in that specific order. The trajectories depicted in Figure 11 effectively demonstrate how our methods enable the robot to smoothly and efficiently reach the desired zones in the prescribed order. These trajectories showcase that our trained agent can efficiently adjust the robot's state after completing a sub-task and promptly proceed toward the next goal zone.

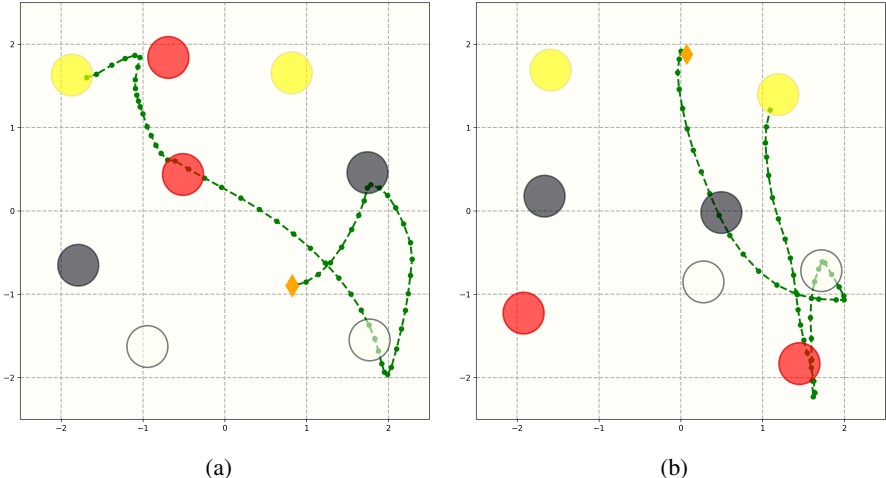

Figure 11: Figures (a) and (b) depict trajectories that solve the goal chaining task: $\mathbf{F}(j \wedge \mathbf{F}(w \wedge \mathbf{F}(r \wedge \mathbf{F}y)))$. The initial positions of the robot are represented by orange diamonds ◆. The different colors of the zones, namely *Red, Yellow, White*, and *Jetblack*, correspond to the task propositions *r, y, w, j*, respectively. It is important to note that the initial positions of the robot and the zones are randomized.

# E  Demonstration on $\omega$-LTL Tasks

## E.1  $\omega$-Loop Tasks

An $\omega$-loop task, expressed in the form $\mathbf{GF}(a \wedge \mathbf{XF}b) \wedge \mathbf{G}(\neg c)$, intuitively requires the agent to continuously loop between two specific zones, $a$ and $b$, while always avoiding another zone, $c$. For example, in the task $\mathbf{GF}(r \wedge \mathbf{XF}y) \wedge \mathbf{G}(\neg w)$, the agent is expected to traverse between the *Red* and *Yellow* zones indefinitely, while never visiting any *White* zone. Figures 12a and 12b illustrate trajectories that solve this $\omega$-LTL task. In both trajectories, the robot quickly reaches the *Red* zone and subsequently oscillates between the *Red* and *Yellow* zones. It is important to note that the LTL formula only requires the avoidance of *White* zones. Therefore, the trajectory shown in Figure 12b, which visits a *Jetblack* zone, is still valid. Theoretically, to satisfy an $\omega$-LTL task, our trained agent should run indefinitely if not falsified. However, in Figure 12, we only plot trajectories for the first 1500 steps to demonstrate that our trained agent is capable of solving $\omega$-loop tasks.

## E.2  $\omega$-Stability Tasks

An $\omega$-stability task requires the agent to satisfy a task proposition by reaching a goal region and remaining in the goal region infinitely. Figures 13a and 13b depict two trajectories that (approximately) satisfy the stability task: $\mathbf{FG}y$. In both figures, the agent can eventually reach a desired *Yellow* zone and then stabilize at that zone. Since the stability task runs infinitely, we report the first 1500 steps to demonstrate that our agent is capable of stabilizing at zones of the specified color.

# F  Demonstrations of the LTL Tasks on Ant 16-room Environments

We conduct experiments to assess the effectiveness of our approach to navigating an Ant through a challenging environment consisting of 16 rooms separated by thick walls. In this environment, each room has the same size $8 \times 8$ divided by walls and corridors with thickness 1 (Fig. 16). There are two obstacles denoted by black squares in the environment. We place a Mujoco (Todorov et al. [2012]) Ant robot in this environment for navigation. To solve the task, the ant should depart from the center of the bottom-left room to reach the desired goal positions. The initial state of the ant is created with random noise, which can keep the position of the ant around the center of the bottom-left room.

Here an atomic proposition $g \in \mathcal{G}$ is in the form of $(r, c)$ denoting the room in the $r$-th row and $c$-th column. The bottom-left corner is room $(0, 0)$. Given an environment state $s$, define $s \vDash (r, c)$ is true if the position of the Mujoco ant in $s$ is close to the center of the $(r, c)$-th room within a threshold.

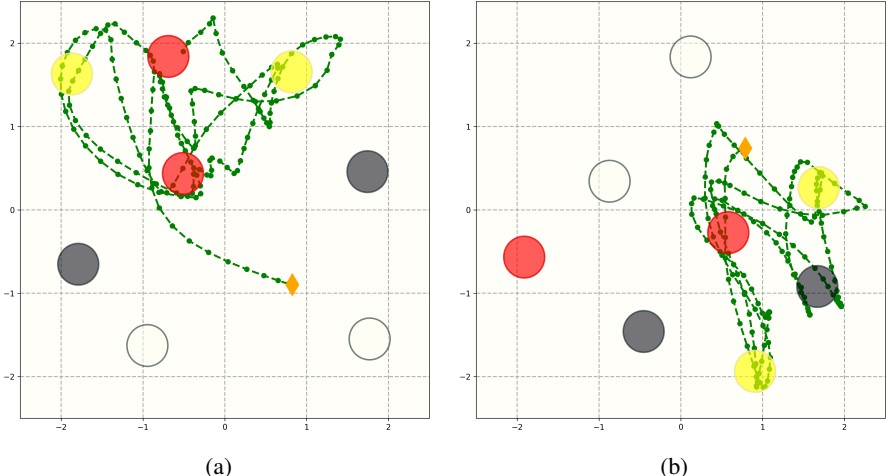

| (a) | (b) |

Figure 12: Figures (a) and (b) illustrate trajectories that solve the $\omega$-loop task: $\mathbf{GF}(r \wedge \mathbf{XF}y) \wedge \mathbf{G}(\neg w)$. The initial positions of the robot are represented by orange diamonds ⬦. The different colors of the zones, namely *Red, Yellow, White*, and *Jetblack*, correspond to the task propositions *r, y, w, j*, respectively. It is important to note that the initial positions of the robot and the zones are randomized.

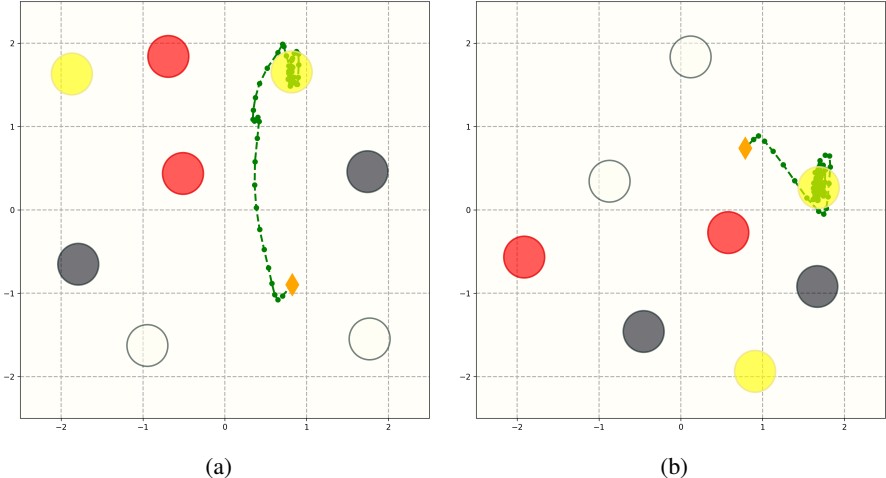

| (a) | (b) |

Figure 13: Figures (a) and (b) illustrate trajectories that solve the $\omega$-stability task: $\mathbf{FG}y$. The initial positions of the robot are represented by orange diamonds ⬦. The different colors of the zones, namely *Red, Yellow, White*, and *Jetblack*, correspond to the task propositions *r, y, w, j*, respectively. It is important to note that the initial positions of the robot and the zones are randomized.

### F.1 Tasks: LTL Specifications

We evaluate our approach using five LTL specifications taken from DiRL Jothimurugan et al. [2021] denoted as $\phi_1$ to $\phi_5$. These tasks have increasing levels of difficulty as they require the agent to sequentially reach a growing number of sub-goals. In order to reach each sub-goal, the ant is allowed a maximum of 1000 steps to move within the environment. In Figure 16, the sub-goals are represented by colors, starting from blue and progressing to orange, yellow, green, and purple for $\phi_1$ to $\phi_5$ respectively. The agent has the option to choose between two possible paths when transitioning from one sub-goal to the next. For instance, in the most complex specification, the agent must depart from the blue point and sequentially reach the sub-goals at orange, yellow, green, and finally purple. The five specifications are described below:

- $\phi_1 := \mathbf{F}\big((0,2) \vee (2,0)\big)$.

It corresponds to reaching the center of either room (0,2) or room (2,0) in the Ant16rooms environment.

- $\phi_2 := \mathbf{F}(((0,2) \vee (2,0)) \wedge \mathbf{F}(2,2))$.

  After completing the task specified by $\phi_1$, the ant should then proceed to reach the center of room (2,2) in the Ant16rooms environment.

- $\phi_3 := \mathbf{F}(((0,2) \vee (2,0)) \wedge \mathbf{F}((2,2) \wedge \mathbf{F}(((2,1) \vee (3,2)) \wedge \mathbf{F}(3,1))))$.

  After completing the task specified by $\phi_2$, the ant should then proceed to reach the center of room (3,1) in the Ant16rooms environment. The ant has the option to go through the center of either room (2,1) or room (3,2) on its way to the target location.

- $\phi_4 := \mathbf{F}(((0,2) \vee (2,0)) \wedge \mathbf{F}((2,2) \wedge \mathbf{F}(((2,1) \vee (3,2)) \wedge \mathbf{F}((3,1) \wedge \mathbf{F}(((1,1) \vee (3,3)) \wedge \mathbf{F}(1,3))))))$.

  After successfully completing the task defined by $\phi_3$, the ant's next objective is to reach the center of room (1,3) in the Ant16rooms environment. The ant has the choice to pass through the center of either room (1,1) or room (3,3) while making its way to the designated location.

- $\phi_5 := \mathbf{F}(((0,2) \vee (2,0)) \wedge \mathbf{F}((2,2) \wedge \mathbf{F}(((2,1) \vee (3,2)) \wedge \mathbf{F}((3,1) \wedge \mathbf{F}(((1,1) \vee (3,3)) \wedge \mathbf{F}((1,3) \wedge \mathbf{F}(((1,1) \vee (0,3)) \wedge \mathbf{F}(0,1))))))))$.

  Upon completing the task specified by $\phi_4$, the ant's subsequent goal is to reach the center of room (0,1) in the Ant16rooms environment. The ant has the option to traverse through the center of either room (1,1) or room (0,3) as it moves towards the target location.

Additionally, to demonstrate the generalization capability of our approach, we incorporate two additional specifications, namely $\phi_6$ and $\phi_7$, which were not included in the original study by DiRL ( Jothimurugan et al. [2021]). Furthermore, we introduce an $\omega$-regular LTL specification, denoted as $\phi_8$, which is not supported by DiRL.

- $\phi_6 := \mathbf{F}(((2,0) \vee (0,1)) \wedge \mathbf{F}((2,1) \wedge \mathbf{F}(((2,2) \vee (1,1)) \wedge \mathbf{F}((1,2) \wedge \mathbf{F}(((3,2) \vee (1,3)) \wedge \mathbf{F}(3,3))))))$.

  It corresponds to the following sequence of sub tasks: First, reaching the center of room (2,1) by passing through either room (2,0) or room (0,1). Next, reaching the center of room (1,2) by passing through either room (2,1) or room (1,1). Finally, reaching the center of room (3,3) by passing through either room (3,2) or room (1,3).

- $\phi_7 := \mathbf{F}(((2,0) \vee (0,2)) \wedge \mathbf{F}((2,2) \wedge \mathbf{F}(((2,1) \vee (1,2)) \wedge \mathbf{F}((1,1) \wedge \mathbf{F}(((0,1) \vee (1,3)) \wedge \mathbf{F}(0,3))))))$.

  It corresponds to the following sequence of sub tasks: First, reaching the center of room (2,2) by passing through either room (2,0) or room (0,2). Next, reaching the center of room (1,1) by passing through either room (2,1) or room (1,2). Finally, reaching the center of room (3,3) by passing through either room (3,1) or room (1,3).

- $\phi_8 := \mathbf{F}(((2,0) \vee (0,2)) \wedge \mathbf{GF}((2,2) \wedge \mathbf{X}(\mathbf{F}((2,1) \wedge \mathbf{XF}(1,1)))))$.

  It corresponds to the following sequence of sub tasks: First, reaching the center of room (2,2) by passing through either room (2,0) or room (0,2). Then, taking an infinite loop from room (2,2) → room (2,1) → room (1,1) → room (1,2) → room (2,2)...

### F.2 Results.

We also compare our approach with DiRL (Jothimurugan et al. [2021]), which is a state-of-the-art LTL-satisfying RL algorithm. Similar to ours, DiRL leverages the compositional structure of a task specification to enable learning. However, there are some differences between our approach and DiRL. In DiRL, a unique low-level policy is learned for each sub-goal transition, and a high-level policy plans over the sub-goal transition space by evaluating the quality of the various low-level policies. One limitation of DiRL is that it is restricted to a small fragment of LTL. Particularly, it cannot handle $\omega$-regular LTL properties that involve infinite state sequences, as it is not possible to learn a unique low-level policy for each transition in an infinite sequence. Additionally, DiRL is not applicable to multi-task RL scenarios because the low-level policies are specific to a single environment setting and do not generalize across different environments.

| Methods | DiRL | DiRL+ GCSL | Ours | Ablation |
|---------|------|------------|------|----------|
| $\phi_1$ | 0.910 (0.022) | 0.923 (0.082) | 0.967 (0.006) | 0.722 (0.073) |
| $\phi_2$ | 0.770 (0.083) | 0.953 (0.037) | 0.925 (0.049) | 0.713 (0.046) |
| $\phi_3$ | 0.367 (0.147) | 0.967 (0.017) | 0.935 (0.028) | 0.667 (0.039) |
| $\phi_4$ | 0.183 (0.046) | 0.937 (0.031) | 0.875 (0.041) | 0.593 (0.086) |
| $\phi_5$ | 0.043 (0.061) | 0.913 (0.017) | 0.868 (0.038) | 0.533 (0.083) |
| $\phi_6$ | / | / | 0.857 (0.004) | 0.565 (0.045) |
| $\phi_7$ | / | / | 0.882 (0.018) | 0.617 (0.043) |
| $\phi_8$ | / | / | 0.903 (0.045) | 0.655 (0.039) |

Table 2: The success rate of the LTL specifications $\phi_1$ to $\phi_8$ on Ant16rooms. The success rates shown are the average values obtained from three individual training experiments (for each training experiment, we evaluate the final trained agent against a specification 200 times) Since **DiRL** and **DiRL+GCSL** cannot be applied to multi-task RL, we train a separate agent for each of the LTL specifications $\phi_1$ to $\phi_5$ using 3e6 environment steps. In contrast, our algorithm trains a single goal-conditioned agent using Algorithm 2 via 3e6 environment steps and evaluates this agent over all eight LTL specifications.

In our experiment, we consider both **DiRL** and **DiRL+GCSL** as our baselines. **DiRL** uses the Augmented Random Search (ARS) algorithm (Mania et al. [2018]) to learn the low-level policy for each sub-goal transition. **DiRL+GCSL**, on the other hand, replaces the ARS learning algorithm with our extended GCSL algorithm (shown in Algorithm 2) with a fixed sub-goal for learning each low-level sub-goal transition policy. This allows for a fair comparison as our method is based on this learning algorithm. Since **DiRL** and **DiRL+GCSL** cannot be applied to multi-task RL, we train a separate agent for each of the LTL specifications $\phi_1$ to $\phi_5$ using 3e6 environment steps. In contrast, our algorithm trains a single goal-conditioned agent using Algorithm 2 via 3e6 environment steps and evaluates this agent over all five LTL specifications.

Table 2 presents the success rates of the baseline approaches **DiRL**, **DiRL+GCSL**, and **Our** approach on all the LTL specifications. The success rates shown are the average values obtained from three individual training experiments (for each training experiment, we evaluate the final trained agent against a specification 200 times). As the complexity of the specifications increases, the performance of **DiRL** gradually deteriorates, while the other baseline approach, **DiRL+GCSL**, maintains a stable success rate across all LTL specifications (from $\phi_1$ to $\phi_5$). Our goal-conditioned agent achieves comparable performance to **DiRL+GCSL**, despite using significantly fewer samples to solve these LTL tasks (see the explanation above). In addition, our approach demonstrates immediate high success rates on the two additional LTL specifications, $\phi_6$ and $\phi_7$, without the need for additional training. In contrast, the **DiRL** baselines require retraining from scratch for these additional tasks. This showcases the capability of our technique to use a simple goal-conditioned RL agent to follow arbitrary LTL specifications without additional training over the LTL task space. Unlike the **DiRL** baselines, which are designed for LTL specifications expressible as regular expressions, our technique is not limited to regular expressions and can handle more complex $\omega$-regular expressions. This is demonstrated by the success rate of our agents on the $\omega$ (loop) specification $\phi_8$, where the agent is evaluated based on its ability to complete the loop twice. To further illustrate the performance of our approach, we provide examples of evaluation rollouts for specifications ranging from $\phi_1$ to $\phi_8$ in Figure 14.

### F.3 Ablation Study

**Random path selection.** Recall that our algorithm uses the goal value function $\mathcal{V}(s_0, g_1, g_2)$ to measure the capability of reaching goal $g_2$ from goal $g_1$ from the viewpoint of the agent at an initial state $s_0$. This information is crucial for performing path planning over the graph representation of a Büchi automaton, enabling the optimal execution of the sequence of tasks defined in an LTL instruction. In this ablation study, we investigate the importance of path planning in our algorithm. We create an ablated version of our approach where acceptable paths are randomly generated from the Büchi automaton of an LTL specification, and the goal-conditioned agent executes towards the goals along these randomly selected paths. The success rates for the ablated version of our algorithm across all specifications are presented in the last column of Table 2, and the example trajectories of the agent for $\phi_1$ to $\phi_8$ in this ablated setting are depicted in Figure 15. Our experiment reveals that without the path planning guided by the goal value function $\mathcal{V}$, the performance in satisfying specifications $\phi_1$ to

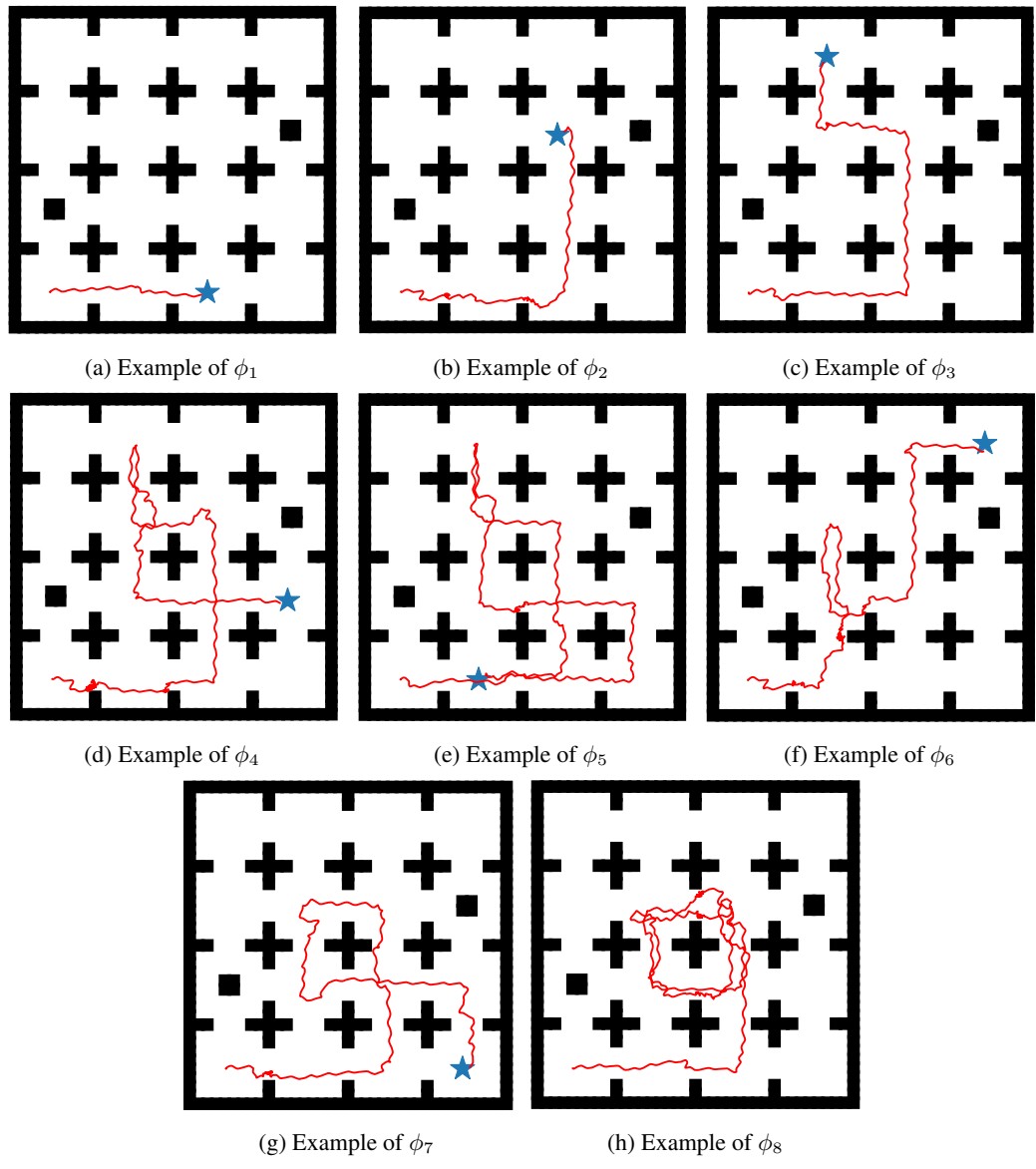

(a) Example of $\phi_1$       (b) Example of $\phi_2$       (c) Example of $\phi_3$

(d) Example of $\phi_4$       (e) Example of $\phi_5$       (f) Example of $\phi_6$

(g) Example of $\phi_7$       (h) Example of $\phi_8$

Figure 14: Figures (a) to (e) depict examples of agent trajectories for each specification from $\phi_1$ to $\phi_8$. In these figures, the ★ symbol represents the final goal position required by the respective specifications. Notably, for $\phi_8$, which is an $\omega$-regular LTL specification involving infinite state sequences, there is no final goal for this specification.

$\phi_8$ significantly deteriorates. This decline can be attributed to the randomly selected paths sometimes crossing obstacles in certain rooms, making it challenging for the Mujoco ant to navigate through them. By comparing Figure 14 (which showcases our full algorithm with path planning) with Figure 15, we observe that our algorithm's ability to select obstacle-free paths (thanks to the goal value function $\mathcal{V}$) significantly improves the agent's capacity to satisfy the given specifications.

## G Implementation Details

### G.1 Environment Details

**ZoneEnv.** The ZoneEnv environment is derived from OpenAI's Safety Gym (Ray et al. [2019]), and it features a continuous action and observation space. The square-shaped walled environment

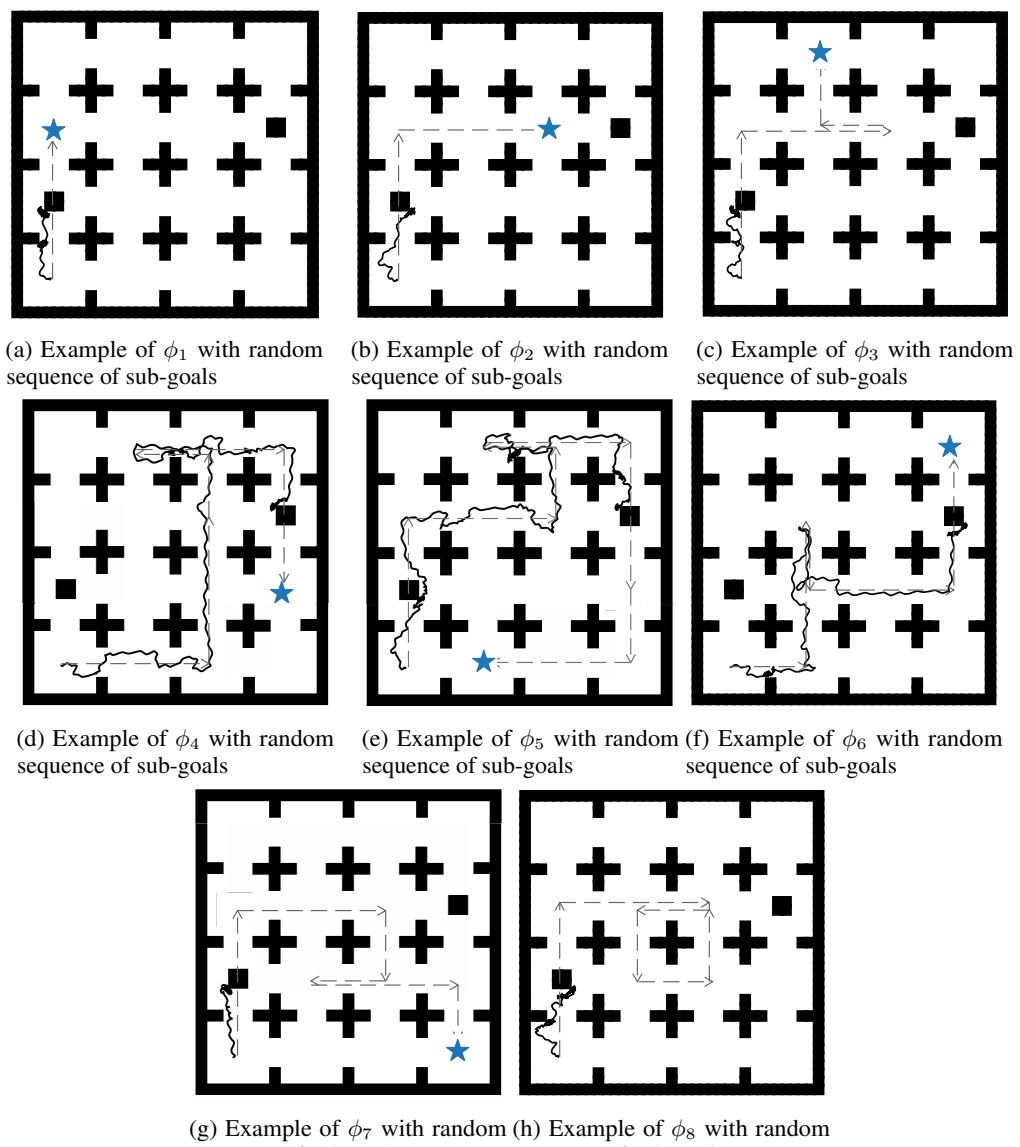

(a) Example of $\phi_1$ with random sequence of sub-goals

(b) Example of $\phi_2$ with random sequence of sub-goals

(c) Example of $\phi_3$ with random sequence of sub-goals

(d) Example of $\phi_4$ with random sequence of sub-goals

(e) Example of $\phi_5$ with random sequence of sub-goals

(f) Example of $\phi_6$ with random sequence of sub-goals

(g) Example of $\phi_7$ with random sequence of sub-goals

(h) Example of $\phi_8$ with random sequence of sub-goals

Figure 15: The examples of failed trajectories of each specification for the abated version of our algorithm. In these figures, the ★ symbol represents the final goal position required by the respective specifications. Randomly selected paths sometimes cross obstacles in certain rooms, making it challenging for the Mujoco ant to navigate through them.

has 8 *zones* (2 of each color). The colors correspond to the task propositions. We use a simple robot from Safety Gym called Point, with one actuator for turning and another for moving forward or backward. An agent can observe the LiDAR information of its surrounding zones. Given this indirect geographical information, it has to visit and/or avoid certain zones to satisfy sampled LTL task specifications. To acquire agents to generalize to any task, the initial positions of zones and the robot are random in every episode. Within this environment, the robot has the ability to perceive its own state, including factors such as velocity, acceleration, and LiDAR information. The observation comprises four distinct groups of LiDAR information, each corresponding to a specific color.

To capture LiDAR information for each color, the environment employs a set of 16 hypothetical light bins emitted from the robot. These bins are evenly distributed, with intervals of 30 degrees between them. Each bin is capable of detecting the presence of a desired color zone in its direction and returns a normalized distance value indicating the proximity of the zone. The returned distance values range

from 0 to 1, with lower values indicating closer zones, 0 denoting overlap between the agent and a specific color zone, satisfying the corresponding proposition, and 1 indicating the absence of a desired zone in that direction. In cases where a bin can intersect with two zones simultaneously, the returned value is determined by the zone that is closer to the robot. The state representation in the `ZoneEnv` environment employs a 76-dimensional vector, consisting of 12 dimensions for the robot and 64 dimensions for the LiDAR information (16 dimensions per color).

In this environment, the goal-labeling function $L$ acts as an event detector, providing a truth assignment for all propositions related to the four colors $\mathcal{G} = j, w, r, y$. The goal-labeling function triggers when the propositions in $\mathcal{G}$ hold true in the environment. For example, $r \in L(s)$ if and only if the agent is located on top of a red zone at a specific environment state $s$.

**Ant-16rooms.** This environment with continuous observation and action space is adapted from the 16 rooms environment from Jothimurugan et al. [2021]. In this walled environment with 16 rooms, each room has the same size $8 \times 8$ divided by walls and corridors with thickness 1 (Fig. 16). There are two obstacles denoted by black squares in the environment. We place a Mujoco (Todorov et al. [2012]) Ant robot in this environment for navigation. To solve the task, the ant should depart from the center of the bottom-left room to reach the desired goal positions. The initial state of the ant is created with random noise, which can keep the position of the ant around the center of the bottom-left room.

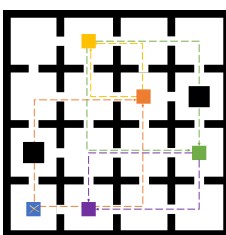

Figure 16: Ant 16 rooms

**LetterEnv.** The `LetterEnv` is a grid-based environment with a size of $7 \times 7$, featuring discrete action and observation spaces. Within this environment, the agent has a complete view of the grid, including its own position. Specifically, for each type of letter, there is a $7 \times 7$ matrix that contains multiple integers. A value of 0 in a grid cell indicates that the letter is not present at that position, while a value of 1 denotes the presence of a letter instance. In addition to these 12 matrices representing the letters, there is an additional matrix indicating the position of the agent. By considering both the letter positions and the agent's position, the goal-labeling function $L$ can determine which task propositions are satisfied or unsatisfied at each step.

### G.2 Primitive Policies Details

**Dynamic Primitives of ZoneEnv.** In order to obtain a discrete action space for the Point robot in Safety Gym (Ray et al. [2019]), we trained a set of four dynamic primitives in the `ZoneEnv` environment using reinforcement learning. Each dynamic primitive was trained for 0.5 million steps. These four dynamic primitives are responsible for guiding the robot to move in the cardinal directions of *UP, DOWN, LEFT*, and *RIGHT* respectively. The reward function used to train these primitives is defined as follows:

$$r_{zone} = c_v \cdot v_{direction} - c_{per} \cdot |v_{per}| \tag{7}$$

where $v_{direction}$ is the velocity of a given direction, and $v_{per}$ denotes the velocity of the direction that is perpendicular to the given direction. In order to train the *UP* and *DOWN* primitives, $v_{direction} = \pm v_x$ tracks velocities on $\pm x$-axis; similarly, to train the *LEFT* and *RIGHT* policies, we set $v_{direction} = \pm v_y$ to keep track of velocities along $\pm y$-axis. Coefficients $c_v, c_{per}$ are set to be 1 and 0.05, respectively. All dynamic primitives for the `ZoneEnv` environment were trained using the PPO algorithm (Schulman et al. [2017]) implemented in the Stable-Baselines3 RL framework (Raffin et al. [2021]).

**Dynamic Primitives of Ant.** For the MuJoCo ant, we follow Qiu and Zhu [2022] and train a set of four basic primitive policies, i.e., *UP, DOWN, LEFT* and *RIGHT* for 1 million steps for each. The reward function to acquire the above four primitives is defined as:

$$r_{ant} = c_v \cdot v_{direction} + c_h \cdot I(\text{IsHealthy}) - c_a \cdot \|a\|_2 - c_f \cdot \|f_{contact}\|_2 \tag{8}$$

where $v_{direction}$ is the velocity of a given direction, $I(\text{IsHealthy})$ is a Boolean function that determines the health condition of the agent, action $a$ is an 8-dimension vector, and $f_{contact}$ is a vector that encodes the contact force of the agent. In order to train the *UP* and *DOWN* primitives, $v_{direction} = \pm v_x$ tracks velocities on $\pm x$-axis; similarly, to train *LEFT* and *RIGHT* policies, we set $v_{direction} = \pm v_y$ to keep track of velocities along $\pm y$-axis. Coefficients $c_v$, $c_h$, $c_a$ and $c_f$ are set to be 1, 1, 0.5, $5 \times 10^{-4}$, respectively. All dynamic primitives for the MuJoCo ant are trained using the Soft Actor-Critic algorithm (Haarnoja et al. [2018]) implemented in the OpenAI Spinning Up RL framework (Achiam [2018]).

**Training a Goal-Conditioned agent for ZoneEnv.**    We employ the goal-conditioned Proximal Policy Optimization (PPO) algorithm, as shown in Algorithm 1, to train a goal-conditioned agent that can satisfy all task propositions (goals) in the `ZoneEnv` environment, which involves reaching zones of different colors. For each color (*Yellow, Red, White*, and *Jetblack*), we use a random vector of 24 dimensions as the goal representation. During training, we sample the goal colors and correspondingly adapt the reward function and the goal representation of the `ZoneEnv` environment. It is crucial to ensure the synchronization of these elements within each episode. For instance, when instructing the agent to reach a *Red* zone, we use the reward function to detect the *r* proposition and exclusively utilize the appropriate goal representation for *Red*. To achieve a balanced agent that can successfully reach zones of all colors with equal rates, we sample the goal colors uniformly. The reward function for acquiring the goal-conditioned agent is defined as follows:

$$r_{color} = \begin{cases} 1, & \text{satisfied} \\ 0, & \text{unsatisfied} \end{cases} \tag{9}$$

where color denotes a goal (i.e., one of *r, y, w, j*) that the agent must reach in an episode. We employ a sparse reward function that only provides a reward of 1 when the desired color is satisfied, while all other situations receive a reward of zero. We do not penalize the agent for stepping into zones of different colors. Our focus is solely on whether the agent can effectively and efficiently reach the desired color.

**Training a Goal-Conditioned agent for LetterEnv.**    We utilize the goal-conditioned Proximal Policy Optimization (PPO) algorithm, as shown in Algorithm 1, to train a goal-conditioned agent capable of satisfying all task propositions (i.e., reaching all letters) in the `LetterEnv` environment. To represent each letter as a goal, we employ a random matrix with a size of $7 \times 7$. The training process for the goal-conditioned agent follows the same methodology as in the `ZoneEnv` environment to ensure synchronization among goals, reward functions, and goal representations in each episode. The reward function for acquiring the goal-reaching primitive is defined as follows:

$$r_{letter} = \begin{cases} 1, & \text{satisfied} \\ 0, & \text{unsatisfied} \end{cases} \tag{10}$$

where letter denotes one of the goal letter that the agent must reach. We employ a sparse reward function that provides a reward of 1 if the desired goal is achieved and 0 otherwise.

**Training a Goal-Conditioned agent for Ant 16 rooms.**    We apply the goal-conditioned supervised learning algorithm described in Algorithm 2 to train a goal-conditioned agent that can navigate in the `Ant16rooms` environment. The reward function is defined as follows:

$$r_g = \begin{cases} 1, & \text{goal reached} \\ 0, & \text{goal not reached} \end{cases} \tag{11}$$

where $g$ is a randomly sampled goal in the goal space.

### G.3   Handling Transitions

In our graph representation $G_{\mathcal{B}}$ of a Büchi automaton $\mathcal{B}$, a "self-transition" on a node $q$ describes the goal-related constraint $\psi'$ that must be maintained until the agent can transition to a neighbor node of $q$ with the underlying goal-conditioned policy $\pi$. If $q$ is on the planned optimal path, the agent for $B$ needs to use $\pi$ to simultaneously ensure $\psi'$ while transitioning to the target region. In the paper, we

primarily focus on reach avoidance, where $\psi' = \bigwedge_k \neg g_k$ encodes regions in the goal space to avoid. At a current environment state $s$, when the value function $V^\pi(s, g_k)$ is greater than a threshold, we take a safe action $\arg\min_a Q^\pi(s, g_k, a)$ that moves the agent away from the most dangerous zone $g_k$. Similarly, our strategy applies to cases where $\psi' = \bigwedge_k g_k$ encodes goal regions that the agent must stay within before transitioning out from $q$. In such cases, if the value function $V^\pi(s, g_k)$ is less than a threshold, we could take an action $\arg\max_a Q^\pi(s, g_k, a)$ that encourages the agent to remain in $g_k$. As our navigation benchmarks do not support the evaluation of this feature, we plan to explore it in our future work.

In the paper, we restrict the target region for a transition as $\psi = \bigwedge_j g_j$. Our tool supports more diverse target regions in the form of $\psi = \bigwedge_j g_j \wedge \bigwedge_l \neg g_l$ to handle tasks such as $\mathbf{F}g_1 \wedge \neg g_2$. The strategy is to reuse our avoidance strategy to avoid $g_2$ when the agent is deemed close to the goal region $g_1$ i.e. the value function output $V^\pi(s, g_1)$ is above the threshold $\sigma$. Fig. 17 presents an example to solve tasks in the form of $\mathbf{F}g_1 \wedge \neg g_2$.

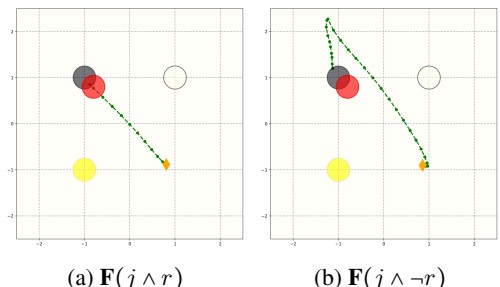

(a) $\mathbf{F}(j \wedge r)$      (b) $\mathbf{F}(j \wedge \neg r)$

Figure 17: Fig (a) and (b) show our capability to handle overlapping goal regions. In (b), the agent takes a detour to avoid touching the red zone.

In the handling avoidance in `ZoneEnv` and `LetterEnv`, finding appropriate value thresholds to *define dangerous* is quite necessary. As we discussed in Sec. G.2, we use sparse reward functions that only yield 0 or 1 in the training process, so the expected threshold to be deemed as *dangerous* should be located somewhere close to (but less than) 1. We perform experiments to study success rates given different value thresholds for both environments. The results are summarized in Table 3. From the table, we can see that a relatively smaller $\sigma$ works better with `ZoneEnv`, while a larger value threshold tends to be a suitable choice for `LetterEnv`. According to the results, we set the $\sigma = 0.85$ for `ZoneEnv`, and $\sigma = 0.92$ for `LetterEnv`.

| Environment | Value threshold $\sigma$ | | | | | |
|---|---|---|---|---|---|---|
| | $\sigma=0.80$ | $\sigma=0.85$ | $\sigma=0.87$ | $\sigma=0.90$ | $\sigma=0.92$ | $\sigma=0.95$ |
| `ZoneEnv` | 0.912 | **0.958** | 0.941 | 0.932 | 0.902 | 0.878 |
| `LetterEnv` | 0.635 | 0.788 | 0.812 | 0.804 | **0.845** | 0.820 |

Table 3: Performance of trained agents on avoidance tasks with different value thresholds $\sigma$. We report the success rates averaged over 1000 episodes.

### G.4 Hyperparameters

The following hyperparameters are used to train the dynamic primitive policies for `ZoneEnv` with PPO (Schulman et al. [2017]).

- Discount factor $\gamma = 0.998$.
- SGD optimizer; actor learning rate 0.001; critic learning rate 0.001.
- Mini-batch size $n = 256$.

The following hyperparameters are used to train the dynamic primitive policies for `LetterEnv` with PPO.

- Discount factor $\gamma = 0.94$.
- SGD optimizer; actor learning rate 0.001; critic learning rate 0.001.
- Mini-batch size $n = 256$.

The following hyperparameters are used to train the goal-conditioned agent for `ZoneEnv` with the goal-conditioned PPO algorithm shown in Algorithm 1.

- Discount factor $\gamma = 0.998$.
- SGD optimizer; actor learning rate 0.0003; critic learning rate 0.0003.
- Mini-batch size $n = 1000$.

The following hyperparameters are used to train the goal-conditioned agent for `LetterEnv` with the goal-conditioned PPO algorithm shown in Algorithm 1.

- Discount factor $\gamma = 0.94$.
- Adam optimizer; actor learning rate 0.001; critic learning rate 0.001.
- Mini-batch size $n = 256$.

The following hyperparameters are used to train the dynamic primitive policies for `Ant` 16 rooms with the SAC (Haarnoja et al. [2018]) algorithm.

- Discount factor $\gamma = 0.99$.
- SGD optimizer; actor learning rate 0.001; critic learning rate 0.001.
- Mini-batch size $n = 100$.
- Replay buffer size 100000.
- Soft update targets $\tau = 0.005$.
- Target update interval and gradient step are set to be 1.

The following hyperparameters are used to train the goal value function $\mathcal{V}$ and policy $\pi$ with our extended goal-conditioned iterative supervised learning algorithm shown in Algorithm 2.

- For goal value function $\mathcal{V}$ :
  - Discount factor $\gamma = 0.99$.
  - Adam optimizer; learning rate 0.001.
  - Mini-batch size $n = 100$.
- For policy $\pi$ :
  - Adam optimizer; learning rate 0.0005.
  - Replay buffer size 20000000.
  - Mini-batch size $n = 512$.

For all training with PPO algorithm, we use GAE $\lambda = 0.97$, clip range $\epsilon = 0.2$, and we set the number of iterations when optimizing the surrogate loss to be 10.

Table 4 summarizes the neural network architectures used by our reinforcement learning algorithms. All neural networks are multi-layer feed-forward neural networks.

| Policies | Environment | Hidden Units |
|---|---|---|
| PPO dynamic | `ZoenEnv` | (64, 64) |
| PPO dynamic | `LetterEnv` | (64, 64) |
| PPO goal-reaching | `ZoneEnv` | (512, 1024, 256) |
| PPO goal-reaching | `LetterEnv` | (256, 512, 128) |
| SAC dynamic | `Ant` | (256, 256) |
| Goal-conitioned value | `Ant16rooms` | (256, 256,256) |
| GCSL | `Ant16rooms` | (256, 256,256) |

Table 4: Neural Network Structures.

