# OpenReview forum: "Instructing Goal-Conditioned Reinforcement Learning Agents with Temporal Logic Objectives"
_NeurIPS.cc/2023/Conference — NeurIPS 2023 poster_

### Official Review · Reviewer_oQVB · 2023-06-13

**Soundness:** 2 fair
**Presentation:** 2 fair
**Contribution:** 3 good
**Rating:** 7
**Confidence:** 4

**Summary:**

This paper considers the problem of instructing goal-conditioned RL agents to follow specifications expressed in Linear Temporal Logic (LTL) formulae. The proposed method works as follows. First, construct a Buchi automaton from the LTL specification, which is then converted to a directed graph representation. Then, use a weighted-graph search algorithm to solve for a high-level plan that satisfies the LTL specification, utilizing the value function of the goal-conditioned agent as a surrogate of the difficulties of achieving each goal. Finally, execute the high-level plan using the goal-conditioned agent. The proposed method is evaluated on three benchmark environments: LetterWorld, ZoneEnv, and Ant-16rooms. The method is compared with two baselines for learning LTL satisfying policies. It is shown to outperform the two baselines, as well as generalize better on out-of-distribution tasks.


**Strengths:**

This paper focuses on a promising direction and targets an important problem of the field: how to learn/search policies that can generalize to complex, compositional task specifications, while using less or no additional training on the new tasks. The problem formulation of LTL specifications is a fruitful step towards this general direction, therefore of great potential significance.

The proposed method of using Buchi automaton and weighted graph search (using value functions as weights to measure difficulty of achieving each goal) to solve high-level plans for LTL tasks is technically interesting and novel to my knowledge. It also makes sense intuitively and seems to be a good method for this problem.

**Weaknesses:**

In my view, there are several improvements that needs to be made in terms of experimental settings and expositions of the paper before it is ready to publish here.

- The proposed method considers the setting where low-level policies to achieve each goal are given, and targets the problem of how to solve for a high-level plan that can satisfy the LTL task specification. In this case, the baselines should be alternative methods for computing a high-level plan, with the same assumption that the low-level policies are given. Then the experiments can show how good the proposed method is in solving the problem it targets. It seems that the current baselines do not operate on the same premise (i.e., given low-level policies for each goal, how to solve high-level plan).
- The writing of the paper could be improved to help with clarity. For example, it would be helpful to briefly introduce how the proposed method works and summarize the experimental results in the introduction section. There are also a few typos in the paper, e.g., line 70 “white zones”.


**Questions:**

Are the baselines also given a goal-conditioned agent, and focusing on solving the high-level planning problem?


**Limitations:**

I did not find a discussion on the limitations in this paper. One possible aspect for discussion is what to do when the task cannot be divided into high-level LTL solving and low-level goal achieving. For example, the case where how to achieve each goal is context dependent.

---

> ### Author Rebuttal · Authors · 2023-08-10
>
> We appreciate your insightful feedback and constructive comments! We present our response to each of your concerns and questions below.
>
> **R1. Comparison with the baselines**: "The baselines should be alternative methods for computing a high-level plan, with the same assumption that the low-level policies are given."
>
> We thank the reviewer for this valuable suggestion. In the supplementary material line 792 Sec E.3, we included a baseline for random high-level path selection, evaluated using the Ant 16-room environment and LTL specifications 1 to 8 (line 700 Sec E.1). Recall that our algorithm uses the goal value function $\mathcal{V}(s_0, g_1, g_2)$ to measure the capability of reaching goal $g_2$ from goal $g_1$ from the viewpoint of the agent at an initial state $s_0$. This information is crucial for performing high-level task planning over the graph representation of a Büchi automaton. The baseline randomly generates high-level paths from the Büchi automaton of an LTL specification, and the goal-conditioned agent executes toward the goals along these randomly selected paths. The success rates for the ablated version of our algorithm across all specifications are presented in the last column of Table 2, and the example trajectories of the agent for LTL specifications 1 to 8 in this ablated setting are depicted in Figure 15. This result reveals that without high-level path planning guided by the goal value function $\mathcal{V}$, the performance in satisfying LTL specifications 1 to 8 significantly deteriorates. This decline can be attributed to the randomly selected paths sometimes crossing obstacles in certain rooms, making it challenging for the Mujoco ant to navigate through them.
>
> During the rebuttal period, we additionally compared our approach with the Logical Options Framework (LQF) [1] baseline. The LQF baseline learns a meta-policy for choosing amongst the options to reach subgoals to reach the final state of the finite state automaton representation of an LTL property. The learning algorithm is based on value iteration over the product of the finite automaton and an environment. That is on every step in the environment, two transitions are applied: the option transition and the finite state automation transition. The options can be recombined to fulfill new tasks. Compared to our technique, LQF only supports co-safe LTL [2] where the "always" operator is not allowed and the "next", "until", and "eventually" operators can only be used in positive normal form. We compared our technique with LQF given the same goal-condition policy for subgoals using the Ant 16-room environment and LTL specifications 1 to 7 (line 700 Sec E.1). We excluded specification 8 because it is not supported by LQF.  We find the two strategies learned the same high-level planning strategy but LQF takes ∼50-100 retraining steps, while our technique generalizes zero-shot to these specifications and we additionally support the $\omega$-regular specification 8. We will include these additional results in a revised version of our paper.
>
> We did not compare our approach with LQF using the colored ZoneEnv environment because this is a multi-task benchmark, which is out of the capability of the meta-policy learning algorithm in LQF. We are unaware of any existing high-level planning algorithms for LTL that generalize to multi-task and out-of-training-distribution environments in a zero-shot manner. To our best, our technique is the first algorithm that can do so.
>
> [1] Brandon Araki, Xiao Li, Kiran Vodrahalli, Jonathan A. DeCastro, Micah J. Fry, Daniela Rus. The Logical Options Framework. ICML 2021: 307-317
> [2] Amit Bhatia, Lydia E. Kavraki, Moshe Y. Vardi. Sampling-based motion planning with temporal goals. ICRA 2010: 2689-2696
>
> **R2. The writing of the paper could be improved to help with clarity**
>
> We appreciate your suggestion for improving the writing of our paper. We will introduce how the proposed method works at a high level and summarize the experimental results in the introduction section, and we will correct the typos in the paper.
>
> **R3. Are the baselines also given a goal-conditioned agent, and focusing on solving the high-level planning problem?**
>
> Please see R1 above.
>
> **R4. Limitations:**
>
> We thank the reviewer for the suggestion to explicitly discuss the limitations of our approach. Indeed, we assume the atomic propositions in LTL properties *can only be goals* within the goal space of goal-condition policies e.g. colored zones in the ZoneEnv navigation benchmark. We do not allow other sources of atomic propositions e.g. external environment signals that are out of the agent's control. For example, our current algorithm does not apply when the agent needs to pursue different tasks based on an external signal. The reviewer is indeed correct in pointing out that our current strategy may not be sufficient when achieving each goal depends on context-dependent external environment signals. We will clarify this limitation in a revised version of the paper. Please also refer to our global response for a justification of our current strategy.
>
> A potential solution to the aforementioned limitation is using a task monitor, which acts as an external memory, to maintain a record of completed sub-goals and past external environment signals. During task execution, when receiving a new environmental signal, our task planning algorithm can dynamically revise the high-level path for the remaining sub-goals that the goal-conditioned agent needs to achieve. We leave it for future work.

---

> > ### Comment · Reviewer_oQVB · 2023-08-13
> > **Re: Rebuttal**
> >
> > Thank you for the detailed rebuttal. The new experiments with baselines that learn high-level plans given low-level policies addressed my concerns and made the paper more complete and solid. I am happy to revise my score.

---

> > > ### Author Response · Authors · 2023-08-16
> > > **Thank you!**
> > >
> > > We are grateful to the reviewer for taking into account our rebuttal and the new experimental results. We will integrate the results and the discussion into the main paper.

---

### Official Review · Reviewer_szt3 · 2023-07-03

**Soundness:** 2 fair
**Presentation:** 2 fair
**Contribution:** 2 fair
**Rating:** 5
**Confidence:** 4

**Summary:**

The paper proposes a new technique for multi-task RL when the tasks are specified using a high-level language (LTL in this case). The approach involves identifying a set of skills corresponding to a set of reachability and safety objectives and training policies for them. While training these policies (which are represented using a single goal-conditioned policy), a separate value function is trained to measure, for every pair of goals, the expected return for the task of reaching one goal from another. Then, given an LTL formula, a subtask graph structure is constructed which is used to compute a high-level plan for performing the task using the learned skills. Experimental results suggest that the proposed approach outperforms a state-of-the-art method for multi-task RL (for LTL tasks) and is better suited for multi-task performance than another compositional approach which is designed for single-task RL.

**Strengths:**

- The ability to learn a set of skills that can be used to perform a wide range of long-horizon tasks specified using LTL is very useful. This proposed approach is a simple and natural way to achieve this.

- Although the idea of planning over a graph structure in order to perform a complex temporal task is not new, the paper provides a way to achieve this for all of LTL (rather than a subset of LTL considered in prior work) and furthermore applies the idea to multi-task RL.

- The experimental results look promising and show that the proposed approach can be used in a wide range of environments to solve complex tasks without further training (after training the goal-conditioned policy)

**Weaknesses:**

- The main weakness, in my opinion, is that the approach doesn't seem to be general enough to handle all of LTL as claimed by the authors. For instance, the LTL task is eventually reduced to following a single path (with a loop at the end) in the automaton graph. But it might not be optimal to follow a single path and one might have to use different high-level strategies from different states of the MDP. Furthermore, the high-level plan is computed using the trained value function which does not consider the ability to stay safe and avoid triggering alternate transitions when measuring the ability to trigger a specific transition in the automaton. The heuristics for handling such avoidance constraints during test time seems reasonable but it is a bit ad-hoc and it is unclear why it is good in general (an ablation study might improve the paper).
- The clarity of the paper could be improved. Many assumptions are made throughout the paper (such as transitions using conjunctive predicates and goals being disjoint). It appears that some of these assumptions can be removed. A clearer presentation could help mitigate doubts about what assumptions are necessary.

**Questions:**

- It is mentioned that disjunctions can be handled by adding separate transitions. But conjunctions seem to be restrictive too. For example, the assumption doesn't allow for a transition with predicate $a\land\lnot b$. How are such things handled?
- How is the cycle to loop within an accepting SCC picked after reaching an accepting state?
- How are the goals (predicates) chosen for training the goal-conditioned policy during an episode?
- Is there a class of MDPs or LTL formulas for which the proposed approach of computing a single path in the automaton can be justified?

**Limitations:**

As mentioned in weaknesses, there are some limitations to the proposed approach which should be discussed in the paper.

---

> ### Author Rebuttal · Authors · 2023-08-10
>
> We appreciate your insightful feedback and constructive comments! We present our response to each of your concerns and questions below.
>
> **R1. Single path problem:** "The LTL task is eventually reduced to following a single path in the automaton graph. But one might have to use different high-level strategies from different states of the MDP."
>
> We clarify that our task planning algorithm may choose different high-level paths for the same LTL task, contingent upon the specific initial environment states. For example, consider the Ant 16-room navigation environments shown in Fig (m) and Fig (n) in the global rebuttal. Based on the learned value function (formalized in line 185), for different initial environment states, our task planning algorithm chooses distinct high-level paths tailored for these initial states.
>
> **R2. What are our assumptions?**
>
> We assume the atomic propositions in LTL properties *can only be goals* within the goal space of the underlying goal-conditioned policy e.g. colored zones in ZoneEnv navigation. We do not allow other sources of atomic propositions e.g. external environment signals that are out of the agent's control. For example, our current algorithm does not apply when the agent needs to pursue different tasks based on an external signal. We will clarify this assumption.
>
> *This limitation is related to the *single path problem* raised in R1.* A potential solution is using a task monitor, which acts as an external memory, to maintain a record of completed sub-goals and past external environment signals. During task execution, when receiving a new environmental signal, our task planning algorithm can dynamically revise the high-level path for the remaining sub-goals that the goal-conditioned agent needs to achieve. We leave it for future work.
>
> **R3. Handling avoidance constraints:** "The high-level plan is computed using the trained value function which does not consider the ability to stay safe and avoid triggering alternate transitions. The heuristics for handling such avoidance constraints during test time seems reasonable but it is a bit ad-hoc and it is unclear why it is good in general (an ablation study might improve the paper)."
>
> We acknowledge the reviewer's valid point regarding the limitation of our task-planning approach, which does not consider the goal-conditioned agent's ability to stay safe before reaching a sub-goal. One solution is to learn an *extended* value function as described in [1]. A Boolean composition of the extended value functions for goal reaching and staying away from unsafe zones can provide a more accurate estimation of the agent's capability of triggering a specific transition in the automaton. We leave the integration with [1] for future work and will clarify this limitation.
>
> We indeed provided an ablation study for handling avoidance at test time. The results were included in the supplementary material line 924, Sec F.3. We investigated the impact of the value threshold $\sigma$ for avoidance. Given a goal-conditioned policy $\pi$, If the value function $V^\pi(s, g) \ge \sigma$ at an environment state $s$ (implying the agent is close to a region $g$ in the goal space to avoid), the agent must take safe actions to move away from $g$ (line 908). As we used sparse reward functions that yield only 0 or 1 for training $\pi$, $\sigma$ should be set somewhere close to (but less than) 1. The results were summarized in Table 3, where we observe that an excessively large value of $\sigma$ compromises the agent's ability for goal reaching, while an insufficiently small value negatively impacts the agent's ability to stay safe.
>
> [1] Nangue Tasse, G., James, S. and Rosman, B. A boolean task algebra for reinforcement learning. NeurIPS 2020.
>
> **R4. How do we handle conjunctive predicates for a transition?**
>
> Our strategy supports environments with overlapping subgoal regions (line 919 in the supplementary material). Given a goal-conditioned policy $\pi$, we support $F(g_1 \wedge g_2)$ by at any environemnt state $s$ taking an action $\arg\max_a{min(Q^\pi(s, g_1, a), Q^\pi(s, g_2, a))}$. We support $F(g_1 \wedge \neg g_2)$ by reusing our avoidance strategy to avoid $g_2$ when the agent is deemed close to the goal region $g_1$ i.e. the value function output $V^\pi(s, g_1)$ is above the threshold $\sigma$. Please see an extended discussion on our avoidance strategy in line 896, Sec F.3. Fig (g) and (h) in the global rebuttal provide examples demonstrating how conjunctive predicates are handled for transitioning. Compared with the path in Fig (g), the agent in Fig (h) takes a detour to avoid touching the red zone.
>
> **R5. How is the cycle to loop within an accepting SCC picked after reaching an accepting state?**
>
> After reaching an accepting state, we apply Dijkstra's algorithm to find the shortest cycle in the SCC that contains the accepting state. Edge weights are determined by the trained value function that measures, for every pair of goals, the expected return of reaching one goal from another.
>
> **R6. How are the goals chosen for training the goal-conditioned policy during an episode?**
>
> Our technique assumes the existence of a goal-labeling function $L$ that maps environment states to valid goals. The goals in ZoneEnv are colored zones: Yellow $y$, Red $r$, White $w$, and Jetblack $j$. For example, $r \in L(s)$ if and only if the agent steps onto a red zone at an environment state s. For each color, we use a fixed random vector as the goal representation for the goal-conditioned policy. The goals in Ant 16 rooms are in the form of $(r, c)$ that denotes the horizontal and vertical (integer) positions of the Mujoco ant. Given an environment state $s$, define $L(s) = (r, c)$ if the position of the ant in $s$ is close to $(r, c)$ within a threshold. During training, we randomly sample initial states and goals and correspondingly adapt the reward function and the goal representation. More details are in the supplementary material Sec E.

---

> > ### Author Response · Authors · 2023-08-16
> > **Single Path Problem**
> >
> > We thank the reviewer again for the insightful comments. We revisited the single-path problem and conducted further experiments during the rebuttal period.
> >
> > > The LTL task is eventually reduced to following a single path in the automaton graph. But one might have to use different high-level strategies from different states of the MDP.
> >
> > We experimented with an alternative high-level planning strategy to evaluate the necessity of different high-level strategies from different states of the MDP *in our context*, using the Ant 16-room environment. We applied LTL specifications 1 to 8 (line 700 Sec E.1) to this environment. Specifically, we employed the goal-conditioned policy to follow a high-level plan, attempting the next sub-goal in the shortest path (in our graph representation of the LTL specification) for $h$ time steps, and subsequently replanning from the current environment state $s_t$ for the remaining sub-goals every $h$ time steps. Before replanning, we updated the weight of any edge between two sub-goals $g_1$ and $g_2$ based on $s_t$ and the trained goal value function $\mathcal{V}(s_t, g_1, g_2)$. We tried various values of $h$. However, the results do not exhibit significant differences compared to our original setting without replanning. For instance, considering the most complex LTL specification $\phi_5$ (line 729), our average success rate over 3 trained policies (each evaluated 200 times) is 86.8% without replanning (Table 2), while with replanning ($h=50$), the success rate is 84.7%.
> >
> > In the experiments, our approach doesn't seem to rely on replanning (and different high-level strategies from different environment states) mainly because the high-level graph structure of a temporal task is already provided in its LTL specification and hence known to us. Prior approaches that combine learning and planning such as [1,2] require iterative replanning due to a lack of knowledge about temporal task specifications and thus have to *sample* the task's high-level graph structure for planning purposes. It's worth noting that our approach could be combined with [1] for improved sub-goal reaching. As illustrated in the rebuttal, integrating our technique with iterative planning can also address the current limitation of handling external environment signals (see R2). We will include the above discussion in our paper.
> >
> > [1] Ben Eysenbach, Ruslan Salakhutdinov, Sergey Levine. Search on the Replay Buffer: Bridging Planning and Reinforcement Learning. NeurIPS 2019
> >
> > [2] Kara Liu, Thanard Kurutach, Christine Tung, Pieter Abbeel, Aviv Tamar. Hallucinative Topological Memory for Zero-Shot Visual Planning. ICML 2020

---

> > > ### Comment · Reviewer_szt3 · 2023-08-19
> > >
> > > Thanks for the detailed response. The additional experiments and the rebuttal helped mitigate some of my concerns. I am revising my score and would encourage the authors to add the clarifications provided in the rebuttal to the paper.

---

> > > > ### Author Response · Authors · 2023-08-21
> > > > **Thank you!**
> > > >
> > > > We appreciate the reviewer's consideration of our rebuttal and the new experiment results. We will incorporate these results and the corresponding discussion into the main paper.

---

### Official Review · Reviewer_Q7B2 · 2023-07-06

**Soundness:** 3 good
**Presentation:** 3 good
**Contribution:** 4 excellent
**Rating:** 8
**Confidence:** 4

**Summary:**

This paper presents a method to transfer learned or planned goal-directed skills in domains to novel tasks represented by linear temporal logic within the same domain. The key idea of this paper is to train goal conditioned policies to achieve (and avoid) Boolean goals, and to compose them temporally to achieve temporal logic goals.

The paper proposes to first convert the automaton corresponding to the LTL specification into a Buchi automaton, which is then converted into a directed graph with a target state. The algorithm also estimates the cost-to-go heuristic at each node to estimate edge traversal costs, and finally combines an optimal graph search along with learned goal conditioned policies to achieve the temporal logic goals.

The authors primarily benchmark against LTL2Action where the specification is embedded into a feature vector using a graph neural network, and this embedded latent feature vector is used alongside a state-feature vector in a through a Deep-RL algorithm to compute the final policy. The authors demonstrate that their approach beats LTL2Action on range of randomly sampled tasks, and two specific avoidance tasks.

**Strengths:**

**Sound problem definition**: The authors are correct in their statement that with competent goal-conditioned policies available to the agent, the agent can solve temporal LTL tasks through composition of these policies. There has been a lot of recent interest in this approach, and the authors have demonstrated that in two navigational environments that have been utilized in research on RL + temporal goals. There are however some issues with the assumptions made by the authors as I describe in the following sections.

**Evaluations**: The authors might have chosen just two navigational domains, but have focused on evaluating over a wide range of temporal logic formulas. Such evaluations are much more valuable in the space of RL for temporal tasks as is considered in this paper.

**Originality and significance**: Prior work suffers from lack of transferrability to all novel tasks as the library of learned skills is inadequate to transfer to all possible tasks. The authors propose to pre-train a goal conditioned skills that should offer more coverage of the logical transition-space. The idea is well demonstrated in the zones and letters environment in the paper, but there are some additional concerns that I highlight in the next section.

Overall, the idea of composing pre-learned skills to novel task scenarios is not original in and of itself. But the combination presented in this paper is novel to the best of my knowledge. However, there are elements of similarity with prior work that have not been addressed adequately

**Weaknesses:**

**Positioning in context of prior work**: The core idea of skill reuse is not entirely novel. There are prior works addressing this [1],[2],[3] that appear to be missing from discussion. Infact both these works handle a wider variety of logical composition for the transition edges which appear to not be considered by this paper. While the core idea in this paper is distinct, it deserves to be discussed in context of these works that appear to handle the problem with greater generality.

**Correctness concerns**: The authors claim that their approach is applicable to all $\omega$-regular automata. However their approach appears to have a strong reliance on a single unique accepting state, in all the examples that they test on. Generalized Buchi acceptance condition requires that the system visit atleast one state in each accepting set infinitely often, and the approach described here is incompatible with such a specification. An example of this would be the patrolling task $\square \diamond a \wedge \square \diamond b$. Here an accepting run would require the agent to visit both $a$, and $b$ infinitely often, but this would not be discoverable by the graph search algorithm described here.

A second correctness concern relates to the type of edge transitions that can be accomplished by the goal conditioned policy. In general an edge transition in automata is described by the self-transition edge that is maintained until the transition trigerring truth evaluation is reached. The goal conditioned policy implicitly assumes that the trigger transition is reached as the first distinct transition in the truth values of the propositions. This might be true for navigational tasks, where each state has at most one proposition true, but may not be true in general when not all propositions are controlled by the agent, or even in cases where simultaneous satisfaction of multiple propositions might be required. For instance the specification $\diamond(a \wedge b)$. [2] would appropriately identify this specification as unsatisfiable, and abort task execution, whereas the behavior of the system proposed in this paper is uncertain. In particular, none of the goal conditioned policies are applicable to this outcome. In contrast [1] can compose the policies logically to satisfy the specification if it is indeed satisfiable. If we study the type of transitions occuring within automata, then there are many such edge cases pertaining to self-transition, and simultaneous truth value changes that cannot be handled by this approach. These limitations must be explicitly acknowledged in the submission.

**Difficulty of training goal-conditioned policies**: This approach relies on having a good goal conditioned policy to perform the task. However, this is in general a challenging problem, and I am not aware of any works that have managed to train competent goal conditioned policies beyond grid world domains that achieve good coverage over all possible logical goals. I would appreciate if the authors add some text to evaluate the quality of goal conditioned policies before using them for this algorithm, or point to works that address this issue.

[1] - Nangue Tasse, G., James, S. and Rosman, B., 2020. A boolean task algebra for reinforcement learning. Advances in Neural Information Processing Systems, 33, pp.9497-9507.

[2] - Liu, J.X., Shah, A., Rosen, E., Konidaris, G. and Tellex, S., 2022. Skill transfer for temporally-extended task specifications. arXiv preprint arXiv:2206.05096.

[3] - Xu, D. and Fekri, F., 2022. Generalizing LTL Instructions via Future Dependent Options. arXiv preprint arXiv:2212.04576.

**Questions:**

These questions pertain to the limitations of the approach proposed, please clarify if your approach can handle it, or update the limitations to include them:
1. Can this approach handle simultaneity? for example ensure goals $a$ and $b$ are true at the same time instant as in $\diamond (a \wedge b)$
2. Can this approach handle self transition that requires maintaining some proposition as true until a goal is reached? For example $\diamond b \wedge a U b$
3. Can the graph search be adopted for recurrence type formulas from Manna and Pnueli's temporal hierarchy [4] (TLDR https://spot.lre.epita.fr/hierarchy.html) $\square \diamond a \wedge \square \diamond b$
4. What are the implicit assumptions you are making on the environment for the validity of the proposed approach? Please clarify these explicitly

[4] - Manna, Z. and Pnueli, A., 1990, August. A hierarchy of temporal properties (invited paper, 1989). In Proceedings of the ninth annual ACM symposium on Principles of distributed computing (pp. 377-410).

**Limitations:**

I do not believe that the limitations are adequately identified and acknowledged. Please refer to the weakness and questions section.

---

> ### Author Rebuttal · Authors · 2023-08-10
>
> We appreciate your insightful feedback and constructive comments!
>
> **R1. Positioning in context of prior work**
>
> We appreciate your suggestion to discuss the related work [1,2,3]. In particular, we will credit [1] for inspiring our approach to managing the logical composition of value functions. Briefly, our method differs by supporting $\omega$-regular LTL properties, which are not immediately applicable in these prior works. Our approach provides equivalent expressivity in terms of logical composition for the transition edges compared to these prior works. Please refer to responses R3-R6 below.
>
> **R2. Single Unique Accepting State**
> - "However their approach appears to have a strong reliance on a single unique accepting state. An example of this would be the patrolling task $GF a \wedge GF b$. Here an accepting run would require the agent to visit both $a$, and $b$ infinitely often, but this would not be discoverable by the graph search algorithm described here."
>
> We respectfully clarify that this is a misunderstanding of our technique. A Büchi automaton accepts an input if and only if it passes through an accepting state infinitely many times as it reads the input. Consider the converted graph representation of the Büchi automaton for the LTL task $GF w \wedge GF y$ depicted in Fig (a) of the global rebuttal. While there is just one accepting state, it only accepts trajectories in which the accepting state is reached infinitely often in a loop, meaning that the agent must visit the white zone ($w$) and the yellow zone ($y$) in Fig (b) infinitely often. This kind of infinite looping behavior $(wy)^\omega$ can be handled by our technique with a goal condition agent capable of reaching $w$ and $y$. As illustrated in the global response, we also support multiple accepting states.
>
> **R3. The type of edge transitions that can be accomplished by a goal-conditioned policy.**
> - 3.1 Can we handle simultaneous satisfaction of multiple propositions?
>
> Yes. Our approach supports simultaneous satisfaction such as $F(g_1 \wedge g_2)$. As illustrated in line 919, Sec F.3 in the supplementary material, to use a goal-conditioned policy $\pi$ to reach the overlapped goal space covered by $g_1$ and $g_2$, at any environment state $s$, we take an action $\arg\max_a{min(Q^\pi(s, g_1, a), Q^\pi(s, g_2, a))}$. Fig. (g) in the global rebuttal provides an example.
>
> - 3.2 Can we handle self-transition that requires maintaining some proposition as true until reaching a goal?
>
> Yes. In our graph representation $G_\mathcal{B}$ of a Büchi automaton $\mathcal{B}$, a "self-transition" on a node $q$ describes the goal-related constraint $\phi$ that must be maintained until the agent can transition to a neighbor node of $q$ with the goal condition $\psi$. If $q$ is on the planned optimal path, the agent for $B$ needs to use a goal-conditioned policy $\pi$ to ensure $\phi$ before accomplishing $\psi$. In the paper, we primarily focus on reach avoidance, where $\phi = \bigwedge_k \neg g_k$ encodes regions in the goal space to avoid. At a current environment state $s$, when the value function $V^\pi(s, g_k)$ is greater than a threshold, we take a safe action $\arg\min_a {Q^\pi(s, g_k, a)}$ that moves the agent away from the dangerous zone $g_k$ (see line 908 for the formalization). Fig. (i) in the global rebuttal provides an example.
>
> Dually, our strategy applies to cases where $\phi = \bigwedge_k g_k$ encodes goal regions that the agent must stay within before transitioning out from $q$. In such cases, if the value function $V^\pi(s, g_k)$ is less than a threshold, we can take an action $\arg\max_a {Q^\pi(s, g_k, a)}$ that encourages the agent to remain in $g_k$. As our navigation benchmarks do not support the evaluation of this feature, we plan to explore it in future work.
>
> **R4. Training goal-conditioned policies**
>
> Our training algorithms are illustrated in Sec. A of the supplementary material.  We evaluated our goal-conditioned policies for ZoneEnv and Ant 16 rooms over 1000 rollouts. We randomly sample the initial agent state and the goal to reach in each rollout. The length of a rollout for ZoneEnv is 500 and for Ant 16 room is 1000. The goal-reaching success rate is 97.2% for ZoneEnv and 67.8% for Ant 16 rooms (evaluated over 3 trials of training). A typical failed case in Ant 16 rooms is shown in Fig. (o) in the global rebuttal where the ant gets stuck at wall corners. Fig (p) highlights the usefulness of LTL instructions to guide the agent to explore a detoured path to the goal.
>
> Indeed, goal-conditioned RL has made substantial progress in recent years. We recommend the following paper that surveys state-of-the-art algorithms for training competent goal-conditioned policies in high-dimensional continuous environments:
>
> Minghuan Liu, Menghui Zhu, Weinan Zhang. Goal-Conditioned Reinforcement Learning: Problems and Solutions. CoRR abs/2201.08299 (2022)
>
> **R5. Can this approach handle simultaneity? e.g. $F(a \wedge b)$?**
>
> Yes. Please see R3.1.
>
> **R6. Can this approach handle self transition that requires maintaining some proposition as true until a goal is reached e.g. $F b \wedge a U b$?**
>
> Yes. Please see R3.2.
>
> **R7. Can the graph search be adopted for recurrence type formulas from Manna and Pnueli's temporal hierarchy $(GF a \wedge GF b)$?**
>
> Yes. Please see Fig (b) in the global rebuttal as an example.
>
> **R8. What are the implicit assumptions you are making on the environment?**
>
> We assume the atomic propositions in LTL properties *can only be goals* within the goal space of the underlying goal-conditioned policy e.g. colored zones in the ZoneEnv navigation benchmark. We do not allow other sources of atomic propositions e.g. external environment signals that are out of the agent's control. For example, our current algorithm does not apply when the agent needs to pursue different tasks based on an external signal. We will clarify this important assumption in the paper.

---

> > ### Comment · Reviewer_Q7B2 · 2023-08-10
> > **Thank you for the detailed response**
> >
> > I apologise for missing the relevant sections in the appendix that address the problem of $\omega$-regular specifications beyond the simple ones described in the main text of the paper. I urge the authors to include at least one such example as a part of the main paper. Further I urge the authors to include the discussion of the the prefix $p$ and the looping suffix $q^\omega$ as a part of the main paper. Finally, as authors mentioned [1] can be used to logically compose value functions almost as a plugin, and perhaps can be included in the main paper as well. Solving $F (a \wedge b)$ utilizes one of their composition operators, and the authors have already experimented with such specifications in the appendix. Finally, once the limitations on the subgoal representations of the propositions are included in the paper, my primary soundness concerns are alleviated. To me adressing all of LTL was a major claim of the paper, and a result of major significance, therefore the harsh rating.
> >
> > The limitation of approximate handling of $\omega$-regular properties remains, but the authors are well aware of it from the rebuttals, and have provided examples in the appendix that are real-world relevant. While it may be heuristic, it is still improving the zero-shot transfer capability beyond prior approaches, and therefore makes a significant contribution.
> >
> > One final limitation is that I don't understand what the method would perform if it were faced with an unsatisfiable task specification, e.g. $F (a\wedge b)$ where there are no regions of the state-space where $a$ and $b$ overlap. Does it 'fail gracefully'? This can be added to the appendix if the authors agree that it would be valuable
> >
> > I am happy to update my score to an accept rating, and would also like to advocate for the paper to be accepted.

---

> > > ### Author Response · Authors · 2023-08-11
> > > **Thank you for your support**
> > >
> > > We appreciate your constructive feedback and your support for our paper!
> > >
> > > We experimented with unsatisfiable task specifications. For properties such as $GF (a \wedge b)$, when the goal regions $a$ and $b$ are close but not overlapping, the agent's behavior mirrors that of $GF a \wedge GF b$, oscillating between $a$ and $b$. However, if these goal regions are far apart, our agent does not exhibit good-looking behavior. We will make sure to acknowledge this limitation in our paper and give concrete examples in the appendix.  We will incorporate all of your suggestions into the paper, as we believe this will significantly improve its quality.

---

### Official Review · Reviewer_VFCf · 2023-07-07

**Soundness:** 2 fair
**Presentation:** 2 fair
**Contribution:** 1 poor
**Rating:** 5
**Confidence:** 4

**Summary:**

This paper considers the problem of learning to solve a linear temporal logic
(LTL) tasks in a Markov Decision Process (MDP). Given a fixed Markov Decision
Process, this is done by:

1. Pre-training a goal-conditioned policy to solve a uniform sampling of reach-avoid tasks,
   where goals correspond to atomic propositions.
1. The input LTL sentence is translated into a Buchi automata.
1. The Buchi automata is transformed into a weighted graph.
   - Weights are determined using the value function of the pre-trained policy.
1. A path is generated by solving a sequence of shortest path problems.

The approach is then experimentally validated against the LTL2Action method using
the prior works domain and concept class.


---- update ----

After re-reading and being pointed to Appendix section D, it seems my major concerns are accounted for. What remains is the question of how to incorporate this into the main text. As such I am increasing my score to erring toward accept.

**Strengths:**

The approach tackles an important problem. Namely, learning to solve sparse
tasks represented in a formal specification language defined over infinite runs
of the system.

**Weaknesses:**

1. The proposed approach is ultimately heuristic and is susceptible to being
   "catastrophically myopic." In particular, the greedy sequence of shortest
   path problems is necessarily biased toward solutions that work for finite
   horizons, but says nothing about the infinite time behavior, e.g., the "lassos"
   generated by sequence of shortest path queries. It is not hard to imagine
   constructing an adversarial Buchi automata that uses a sequence of easy
   to reach accepting states to lead the agent into a long term bad position.

1. The paper claims to address infinite horizon specifications, but then
   compares against a regular language benchmark. This undercuts the stated
   motivation. For example, all of the base-line problems have a finite
   accepting prefix, e.g., as opposed to G(x -> F y).

1. The approach should be compared to hierarchical, meta RL, and compositional
   RL works. For example, the graph approach seems very similar to [1] but adapted
   to goal conditioned policies.

[1] Jothimurugan, Kishor, Rajeev Alur, and Osbert Bastani. "A composable specification language for reinforcement learning tasks." Advances in Neural Information Processing Systems 32 (2019).

**Questions:**

1. How does the system generalize to infinite horizon queries?
1. When is the proposed heuristic guaranteed to work vs have arbitrary failures.

**Limitations:**

See weakness 1.

---

> ### Author Rebuttal · Authors · 2023-08-10
>
> We appreciate your insightful feedback and constructive comments! We present our response to each of your concerns and questions below.
>
> **R1. The proposed approach is ultimately heuristic and is susceptible to being "catastrophically myopic." In particular, the greedy sequence of shortest path problems is necessarily biased toward solutions that work for finite horizons, but says nothing about the infinite time behavior, e.g., the "lassos" generated by sequence of shortest path queries.**
>
> As illustrated in the global rebuttal, our task planning algorithm considers both the shortest path $p$ to an accepting state and the shortest cycle $q$ from the accepting state. Please see Fig (j) and (k) in the global rebuttal as an example. For a complex LTL property $\varphi_1 \vee \varphi_2$ for Ant 16-room navigation in Fig (j), our algorithm picks $\varphi_2$ for the goal-conditioned agent to execute in Fig (k) as it deems $\varphi_1$ more costly although the ``lassos" in $\varphi_1$ is closer to the agent's initial position. We acknowledge that using a bounded $\omega$ overapproximates the true probability for optimal path selection $pq^\omega$, which is a limitation. However, we have the flexibility to use arbitrarily large $\omega$ without incurring additional costs in planning.
>
> **R2. When is the proposed heuristic guaranteed to work vs have arbitrary failures?**
>
> Our task planning technique for temporal properties is based on a goal-conditioned agent trained for reachability properties. For an $\omega$-regular LTL property, while our technique has the flexibility to use arbitrarily large $\omega$ in planning a path for LTL-satisfying runs, it can lead to good-looking bad policies, which induce trajectories that eventually fail to reach the desired accepting state but along the way visited the accepting state arbitrarily many times (potentially prolonging until the heat death of the universe).
>
> **R3. The paper claims to address infinite horizon specifications, but then compares against a regular language benchmark. This undercuts the stated motivation**
>
> We included experimental results of evaluating our algorithm against $\omega$-regular LTL specifications in line 663, Sec. D of the supplementary material (as well as Fig. 14d) which is referred to in the submitted paper. We will incorporate these important experimental results in the main body of the revised paper. Please also see the additional evaluation results in Fig. 8 of the global rebuttal.
>
> **R4. The approach should be compared to hierarchical, meta RL, and compositional RL works.**
>
> We compared our work with DiRL [1], a state-of-the-art compositional RL algorithm for LTL properties. We used DiRL instead of SpecRL [2] as suggested by the reviewer because DiRL is a stronger baseline and empirically outperforms SpecRL on our benchmarks. **The experiments and results were illustrated in line 689, Sec. E of the supplementary material.**
>
> The comparison is conducted on the Ant 16-room environment. The agent needs to navigate a Mujoco Ant in 16 rooms separated by thick walls (Fig. 3). We use 8 LTL specifications (line 700 Sec E.1) including 5 properties taken from DiRL. These tasks have increasing levels of difficulty as they require the agent to sequentially reach a growing number of sub-goals.
>
> Similar to ours, DiRL leverages the compositional structure of a task specification to enable learning. However, there are some key differences. First, in DiRL, a unique low-level policy is learned for each sub-goal transition. Second, DiRL does not support $\omega$-regular LTL properties. Lastly, DiRL is not applicable to multi-task RL scenarios because the low-level policies are specific to a single environment setting and do not generalize across different environments. As a result, we trained a separate agent for each of the LTL specifications for DiRL. In contrast, our algorithm trains a single goal-conditioned agent and evaluates this agent over all 8 LTL specifications.
>
> The results in Sec E.2 show that our agent is able to satisfy all specifications with ~90% success rate trained using 3e6 environment steps, whereas the DiRL method has to exercise 3e6 timesteps for each of the specifications to match the success rate of our approach. Our approach demonstrates immediate high success rates on arbitrarily new LTL specifications without the need for additional training. In contrast, DiRL requires retraining from scratch for new tasks. More comprehensive evaluation results are in Sec E.2 and E.3.
>
> We also compared our approach with a state-of-the-art hierarchical RL algorithm, R-AVI [3]. We used the abstract graph of an LTL specification as input to R-AVI. We found that R-AVI does not scale to the complex LTL specifications for Ant 16-room navigation and can only achieve less than ~40% success rate for all of them when trained using 3e6 environment steps.
>
> We did not compare our approach with DiRL and R-AVI using the colored ZoneEnv environment because this is a multi-task benchmark, which is out of the capability of the learning algorithm in DiRL and R-AVI.
>
> We will move the comparison with DiRL and R-AVI to the main paper in a revised version.
>
> [1] Kishor Jothimurugan, Suguman Bansal, Osbert Bastani, and Rajeev Alur. "Compositional reinforcement learning
> 990 from logical specifications." Advances in Neural Information Processing Systems 34 (2021).
>
> [2] Kishor Jothimurugan, Rajeev Alur, and Osbert Bastani. "A composable specification language for reinforcement learning tasks." Advances in Neural Information Processing Systems 32 (2019).
>
> [3] Kishor Jothimurugan, Osbert Bastani, and Rajeev Alur. Abstract value iteration for hierarchical reinforcement learning. International Conference on Artificial Intelligence and Statistics (2021).
>
> **R5. How does the system generalize to infinite horizon queries?**
>
> Please see R1 and the global rebuttal.

---

> > ### Comment · Reviewer_VFCf · 2023-08-14
> > **Reconsidering**
> >
> > Hi,
> >
> > Thank you for the rebuttal. I will look into the appendix as that seems critical to answering my concerns.
> >
> > I would urge the authors to fit this into the main text if possible, as it seems to have been a very common criticism.

---

> > > ### Author Response · Authors · 2023-08-16
> > > **Thank you for reconsidering**
> > >
> > > We appreciate the reviewer for reevaluating our paper. We promise that we will incorporate the important results regarding $\omega$-regular LTL properties, as highlighted in both the appendix and the rebuttal, into the main text.

---

> > > > ### Author Response · Authors · 2023-08-20
> > > > **Further Discussion**
> > > >
> > > > Dear Reviewer,
> > > >
> > > > We deeply appreciate your willingness to reconsider our paper during the discussion period. We would like to kindly ask if the information presented in our appendix and rebuttal sufficiently resolves your concerns. Should you have any additional concerns, we would highly value the chance to address them before the discussion period concludes. We thank the reviewer again for guiding us to improve the quality of our paper.
> > > >
> > > > Best regards,
> > > >
> > > > The Authors

---

> > > > > ### Comment · Reviewer_VFCf · 2023-08-20
> > > > >
> > > > > Apologies for the delay. I do plan on increasing my score from leaning towards rejection towards acceptance.
> > > > > One thing that would greatly me would be to get a sense of the concrete set of edits planned to incorporate the material from the appendix into the main body of work. The first set of reviews among multiple reviewers changed once the relevant section in the appendix was highlighted. This makes me think that the camera ready version having addressed this is crucial.

---

> > > > > > ### Author Response · Authors · 2023-08-21
> > > > > > **Paper Revisions**
> > > > > >
> > > > > > We deeply appreciate the reviewer for reevaluating our paper. We plan to incorporate the material from the appendix and update the main body of the paper as follows:
> > > > > >
> > > > > > + We will add pseudocode of our task-planning algorithm to clearly explain the handling of both the prefix and looping suffix of a high-level path.
> > > > > > + We will include the experimental results of evaluating our algorithm against the $\omega$-regular LTL specifications that are referenced in both the appendix and the rebuttal.
> > > > > > + We will include the experimental results of comparing our algorithm with hierarchical and compositional RL approaches, such as DiRL, R-AVI, and LQF (as elaborated in our response to reviewer oQVB).
> > > > > > + We will illustrate our support for self-transition edges and the conjunction of atomic propositions (including their negations) in LTL properties (as explained in our response to reviewer Q7B2 and szt3).
> > > > > > + We will provide an explanation of why our approach does not rely on iterative planning (as outlined in our response to reviewer szt3).
> > > > > > + We will discuss the limitations of our work: (1) our assumption that atomic propositions in LTL properties can only be goals within the goal space of goal-conditioned policies; (2) the overapproximation of the true optimal probability by our task-planning algorithm for $\omega$-regular LTL properties; (3) dynamic handling of avoidance during test time instead of during planning (as illustrated in our response to reviewer szt3); (4) the lack of an alarm system for unsatisfiable task specifications (as detailed in our response to reviewer Q7B2).
> > > > > > + We will provide a high-level explanation of how the proposed method operates and summarize the experimental results within the introduction section (suggested by reviewer oQVB) and extend our related work section (suggested by reviewer Q7B2).
> > > > > > + To accommodate the above changes, we will move the experiment results for the discrete Letterworld benchmark and some low-level technical details of reducing LTL properties to graph representations to the appendix.
> > > > > >
> > > > > > We are grateful for the reviewers' constructive suggestions to improve our paper!

---

### Author Rebuttal · Authors · 2023-08-10

We greatly appreciate the valuable feedback and suggestions provided by the reviewers! We will begin by addressing the primary concern raised by the majority of the reviewers in the global rebuttal. We will address the concerns of each reviewer in the individual review responses.

### **How do we support $\omega$-regular  LTL properties?**

Our technique can handle $\omega$-regular LTL specifications even though the underlying goal-conditioned agents have never seen such specifications during training time. ***We included experimental results of applying our algorithm to  $\omega$-regular LTL specifications in the supplementary material line 663, Sec. D which is referred to in the submitted paper. We will incorporate these important experimental results in the main body of the paper.***

**1. Summary of the Task Planning Algorithm.** Our technique generates policies for $\omega$-regular LTL specifications based on a goal-conditioned RL agent and a learned goal value function as follows:
* We first convert $\varphi$ to a Büchi automaton $\mathcal{B}$, which is subsequently converted to a graph representation $G_\mathcal{B}$ using the techqniue illustrated in Sec 3.1.
* We associate edges on $G_\mathcal{B}$ with weight $w$ equal to the capability of the goal-conditioned agent to transition between the goals on the source and target nodes, estimated by the learned $\mathcal{V}$ value function. See the formalization of $\mathcal{V}$ in lines 185-188.
* We decompose $G_\mathcal{B}$ into strongly connected components (SCCs) using Tarjan’s algorithm.
* To find an optimal path on $G_\mathcal{B}$ for task execution to satisfy the LTL $\varphi$, we follow these steps:
    * For each accepting state $s_a$ in a maximal SCC, we use Dijkstra's algorithm to find the shortest path from the initial state to $s_a$, denoted as $p$.
    * Next, we apply Dijkstra's algorithm to find the shortest cycle from $s_a$ back to $s_a$ in the maximal SCC, denoted as $q$.
    * The optimal path for the accepting state $s_a$ is $pq^\omega$ where $\omega \rightarrow \infty$ represents the number of times the shortest cycle is executed. The cost of the optimal path is calculated as $w(p) + \omega \cdot w(q)$ where $w(p)$ or $w(q)$ is the sum of the weights of the edges on $p$ or $q$. In the implementation, we use $\omega = 5$ to estimate the path cost.
    * Finally, we select the optimal path on $G_\mathcal{B}$ as the least-cost path from the initial state to any accepting state of $G_\mathcal{B}$.

**2. Examples.** Please see the demonstrations in Fig. 8 of the attached PDF file where Fig (b), (c), (d), (e), (f), (k), (l) are trajectories produced by our trained goal-conditioned agents instructed by the corresponding $\omega$-regular LTL properties in the ZoneEnv and Ant 16-room environments respectively. We conducted 1000 evaluations for each of these $\omega$-regular LTL properties on randomly sampled environments to determine the task success rate. A task run is deemed successful if the loop from the accepting state on the optimal path chosen by our task planning algorithm can be consecutively executed 5 times within 2000 timesteps during our evaluation. Our success rate surpasses 90% for all these properties. ***We included more thorough experimental results in Sec. D of the supplementary material.***

**3. Justification.** The objective of our task planning algorithm is to find a policy $\pi^* = \arg\max_{\pi} \mathbb{E} \tau \sim \pi(\cdot \vert \cdot, \varphi) \left[ \mathbb{1}[\tau \models \varphi] \right]$, which generates the maximum number of LTL-satisfying runs for a given LTL property $\varphi$. Our formalization of goal-condition RL uses a sparse binary reward function (Equation 2) - a reward of 1 is provided only when the specified goal is successfully achieved by the end of a training episode. In this setting, when the discounted factor $\gamma \rightarrow 1$, the weight $w$ for each edge on $G_\mathcal{B}$, which is determined by the learned value function $\mathcal{V}$ (e.g. $w =-\log{\mathcal{V}(\cdot)}$), is inversely proportional to the (lower bound of the) probability of reaching the goal region represented by the target node from that represented by the source node of the edge. As such, graph search in the task planning algorithm seeks the optimal path as the one the agent is most likely to succeed. The main strength of our technique is its ability to adapt a goal-conditioned agent into one capable of achieving arbitrary LTL properties using atomic propositions from the goal space in a zero-shot manner.  We acknowledge that the use of a bounded $\omega = 5$ overapproximates the true optimal probability, which is a limitation (even though we can use arbitrarily large $\omega$ with no additional cost).

---

### Decision · Program_Chairs · 2023-09-21

**Decision:**

Accept (poster)

**Comment:**

The paper presents an approach to learning to achieve temporal logic goals in an environment. The key novelty is that the approach converts the LTL formula into Buchi automata and then leverages the structure of the automata to learn policies that generalize to other goals.

All reviewers had initial concerns with the paper, but all the concerns were addressed in the rebuttal, mostly with content that was already present in the appendix; in particular some additional experiments. The only remaining concern is that some reviewers feel it is important that the paper better incorporate this content into the main body of the paper. The authors have provided a helpful list of changes to the paper that will address the reviewer concerns.